

# Impact of parameter updates on soil moisture assimilation in a 3D heterogeneous hillslope model

Natascha Brandhorst[1] and Insa Neuweiler[1]

[1]Institute of Fluid Mechanics and Environmental Physics in Civil Engineering, University of Hannover, Hannover, Germany

**Correspondence:** Natascha Brandhorst (brandhorst@hydromech.uni-hannover.de)

**Abstract.** Models of variably saturated subsurface flow require knowledge of the soil hydraulic parameters. However, the determination of these parameters in heterogeneous soils is not easily feasible and subject to large uncertainties. As the modeled soil moisture is very sensitive to these parameters, especially the saturated hydraulic conductivity, porosity and the parameters describing the retention and relative permeability functions, it is likewise highly uncertain. Data assimilation can be used to handle and reduce both, state and parameter uncertainty. In this work, we apply the ensemble Kalman filter (EnKF) to a three-dimensional heterogeneous hillslope model and investigate the influence of updating the different soil hydraulic parameters on the accuracy of the estimated soil moisture. We further examine the usage of a simplified layered soil structure instead of the fully resolved heterogeneous soil structure in the ensemble. It is shown that the best estimates are obtained when performing a joint update of porosity and van Genuchten parameters and optionally the saturated hydraulic conductivity. The usage of a simplified soil structure gave decent estimates of spatially averaged soil moisture in combination with parameter updates but led to a failure of the EnKF and very poor soil moisture estimates at non-observed locations.

## 1 Introduction

Numerical models of the unsaturated zone are very sensitive to the soil hydraulic parameters (Vereecken et al., 1992; Christiaens and Feyen, 2001). Baroni et al. (2010) found this sensitivity to be larger for the more complex but widely used models that solve Richards equation compared to simpler conceptual models. Therefore, knowledge of these parameters is crucial for getting reliable model output, e.g. soil moisture or streamflow estimates if the unsaturated zone is part of a hydrological model.

However, quantification of these parameters is very difficult (Kool et al., 1987). Direct measurements offer the most exact determination. As applications for unsaturated zone models often involve large areas with considerable spatial variability, it is impossible to obtain enough measurements for a decent determination of the soil hydraulic parameters at the required scale (Schaap et al., 2001; Baroni et al., 2010). Therefore, the soil hydraulic parameters are commonly estimated by indirect methods in particular by pedotransfer functions (PTFs). These calculate the soil hydraulic parameters based on information of surrogate data which is easier to be measured, such as soil texture. The drawback of PTFs is that usually a large amount of specific input data is required that is often not fully available and they are very empirical which leads to poor estimates with high uncertainty (Schaap et al., 2001; Christiaens and Feyen, 2001; Baroni et al., 2010). Both, Christiaens and Feyen (2001) and Baroni et al.





(2010), who compared different techniques of estimating the soil hydraulic parameters, found that especially the estimates of the saturated hydraulic conductivity are affected by large uncertainties.

Another possibility to quantify the soil hydraulic parameters is to solve the inverse problem, i.e. finding an adequate set of parameters that best reproduces given observations of a quantity of interest, e.g. soil moisture. This is an optimization problem and can be done by applying one of the manifold of existing optimization algorithms. The advantage of this method is that
the optimization can be applied directly for the given initial and boundary conditions. The disadvantage is that the parameter estimation problem is often ill-posed caused by either non-uniqueness or instability (Yeh, 1986). The problem is non-unique when there exist multiple parameter sets that produce a similar value of the objective function in the optimization. This can be caused by correlations among parameters, when changes in one parameter can be compensated by changes in another parameter, or an insufficient number or information content of observations. The problem is unstable when the paramters are
very sensitive to the observed data, such that a small measurement error would cause large changes in the parameter estimates (Kool et al., 1987). The predominant problem when estimating the soil hydraulic parameters is the non-uniqueness due to insufficient observations. Hornung (1983) and Kool et al. (1985), both, conducted column drainage experiments to determine $\alpha$ and $n$ of van Genuchten's model and both encountered non-uniqueness of the parameters. While Hornung (1983) could help the problem by taking into account additional observations of pressure head, Kool et al. (1985) found that the uniqueness of
these parameters depends on the water content coverage of the observations. The difficulty of estimating a subset of the soil hydraulic parameters in such a simple setup under controlled laboratory conditions adumbrates the impossibility to estimate these parameters fairly in the field.

It is thus clear that parameter uncertainty cannot be eliminated and needs to be taken into account in the modeling process. According to Liu and Gupta (2007), when dealing with uncertainty, one has to address three aspects: understanding, quantifi-
cation and reduction of the uncertainty. A tool for that purpose, that has been given increasing intention in the past decades, is data assimilation. It uses a probabilistic approach to quantify model states and reduces the related uncertainty by the integration of observations aiming for improved estimates of the model states. A big advantage of data assimilation is that all sources of uncertainty can be incorporated, e.g. parameter, model or input uncertainty. The disadvantage is that the updated state and parameter estimates can lead to unphysical combinations that may cause numerical issues.
A very popular data assimilation method is the ensemble Kalman filter (EnKF). Evensen (1994) developed it as a modification of the Kalman filter (KF) for the application to nonlinear models. Here, the probabilistic model description is accomplished by using an ensemble of model realizations that are propagated in time according to the nonlinear numerical model. Even though it is derived only for states and parameters that are Gaussian distributed it has been applied successfully to hydrologic models with non-Gaussian pdfs by e.g. Moradkhani et al. (2005b); De Lannoy et al. (2007) and Erdal et al. (2014).
Another data assimilation method used to account for parameter uncertainty is the particle filter. Although the particle filter is better suited for nonlinear models and does not require Gaussian distributions, its high computational demand rather limits its application to conceptual models (Moradkhani et al., 2005a; Salamon and Feyen, 2009) or models with reduced dimensionality (Montzka et al., 2011).



Many studies put their focus on the accurate estimation of the soil hydraulic parameters, like Li and Ren (2011); Shi
et al. (2015); Chaudhuri et al. (2018) and Zha et al. (2019). Even though these studies aim for parameter estimation in one-
dimensional models only, they encountered problems reproducing the true parameter sets. Li and Ren (2011) were able to
achieve good estimates of the saturated hydraulic conductivity $K_s$ and the van Genuchten model parameter $\alpha$, while the esti-
mates of the remaining soil hydraulic parameters were poor. Shi et al. (2015) found that parameter estimates can be improved
by assimilating different types of observations, i.e. soil water content and groundwater level data, but they restricted the es-
timation to three parameters, namely $K_s$ and the van Genuchten parameters $\alpha$ and $n$. In a similar study, Zha et al. (2019)
identified pressure head observations as the most valuable observations for estimating these three parameters. However, in
field applications, such observations are often not available, because the required measurement devices are less robust, and
one has to resort to measurements of soil water content, groundwater level or streamflow. Chaudhuri et al. (2018) were able
to estimate three-dimensional fields of $K_s, \alpha$ and $n$ with an iterative ensemble Kalman filter. Unfortunately, iterative filters are
computationally quite expensive and therefore less suited for large scale models or real-time predictions.

From these studies it can be learned that parameter updates in filters can hardly be used to estimate the soil hydraulic
parameters under field conditions. Regardless, the parameter updates can have a positive impact on the state estimates when
making predictions. This was shown by Montzka et al. (2011); Wu and Margulis (2011) and Brandhorst et al. (2017) for one-
dimensional cases, where fluxes are vertical. However, it is questionable whether this also holds for three-dimensional models
with heterogeneous soils where the augmented state vector approach can mean a significant increase of the computational
burden and the estimated parameter fields are expected to differ clearly from the true paramater fields. In 3D models, the
uncertainty of model parameters is only one part of the question, while the question of heterogeneity of soil parameters and
related uncertainty is an additional part. Yet, 3D fully coupled subsurface models are used more and more (Goderniaux et al.,
2009; Maxwell et al., 2015).

While parameter updates have been applied to land surface models (Shi et al., 2014; Zhang et al., 2017) or conceptual
models (Moradkhani et al., 2005b), its application to physically based three-dimensional subsurface models is still limited. In
fact, the update of parameters regarding groundwater, like transmissivity or saturated hydraulic conductivity, are commonly
included in the parameter update (Hendricks Franssen and Kinzelbach, 2008; Rasmussen et al., 2015). On the other hand, the
soil hydraulic parameters affecting flow in the vadose zone, are often either updated in a simplified manner by the application
of Miller scaling (Bauser et al., 2020) or global calibration coefficients (Shi et al., 2014) or sometimes even excluded from the
updates as they are prone to cause numerical instabilities (Rasmussen et al., 2015).

To the best of our knowledge, there are no studies on how to treat the uncertain soil hydraulic parameters in three-dimensional
heterogeneous subsurface models in data assimilation. The parameters are either excluded entirely from the update or a (sub)set
of the parameters is included whose choice is not further motivated. The effect of updating different combinations of parameters
on the soil moisture estimates is not analyzed. In Brandhorst et al. (2017), the joint update of all sensitive parameters, i.e.
$K_s, \phi, \alpha$ and $n$ in a van Genuchten parameterization, were found to lead to the best predicitions of soil moisture in a one-
dimensional model with either a homogeneous or a layered soil structure. Consequently, in this work, we want to investigate
the effect of updating the soil hydraulic parameters on the soil moisture estimates in a heterogeneous three-dimensional model.





This comprises the question (1) which parameters should be updated and (2) if and how the heterogeneous soil structure
should be accounted for. We look at a small and steep domain where lateral fluxes in the unsaturated zone are expected and
a one-dimensional representation would be too simplified. For this purpose we set up a numerical model of a hillslope with
heterogeneous soil layers. We assimilate synthetic observations of soil moisture obtained from a reference model run using
the ensemble Kalman filter. The observations shall be representative of continuous sensor data one would obtain from field
measurements. The aim of the filter is to make decent predictions of soil moisture from these observations. Therefore, different
combinations of the soil hydraulic parameters are included in the joint update and the results are compared regarding the
accuracy of the soil moisture estimates. The runs are repeated with an ensemble with homogeneous soil layers to investigate the
effect of applying a simplified soil structure. The focus is on flow in the subsurface and the handling of parameter uncertainty.
The coupling to neighboring compartments as well as the usage of real observations introduce additional error sources and
require special treatment, which would impede a thorough analysis of the effect of the parameter updates. For this reason, the
assimilation is performed using synthetic data and parameter uncertainty is the only error source.

The remainder of this paper is structured as follows: The next chapter explicates the governing equations of the hydrological
model and the ensemble Kalman filter. Additionally, details are given on the specific implementation of the EnKF required
for our simulations. Section 3 describes the different scenarios and the EnKF setup. Afterwards, we present the results of the
data assimilation experiments and discuss the influence of the parameter updates. The paper ends with a summary and the
conclusions of this work.

## 2 Methods

### 2.1 Hydrological model

#### 2.1.1 Subsurface flow

The flow problem is solved with the software ParFlow (Kollet and Maxwell, 2006) which models fully coupled subsurface and
overland flow. Variably saturated flow in the subsurface is represented by the mixed form of Richards equation:

$$S(h_p)S_s\frac{\partial h_p}{\partial t} + \frac{\partial (S(h_p)\phi)}{\partial t} = \nabla \cdot (\mathbf{K}(h_p)\nabla(h_p + z)) = Q \tag{1}$$

where $S$ [$-$] is saturation, $h_p$ [L] is pressure head, $S_s$ [L$^{-1}$] is specific storage, $t$ [T] is time, $\phi$ [$-$] is porosity, $\mathbf{K}$ [LT$^{-1}$] is
the unsaturated hydraulic conductivity tensor, $z$ [L] is the geodetic height and $Q$ [T$^{-1}$] is a source or sink term. The specific
storage is here defined as $S_s = \frac{\phi}{V_t}\frac{\partial V_t}{\partial h_p}$ with $V_t$ [L$^3$] being the total volume and assuming that $\frac{\partial \phi}{\partial h_p}$ is negligible.





The van Genuchten-Mualem model (Van Genuchten, 1980) is used to describe the relation between pressure head, saturation and unsaturated hydraulic conductivity:

$$
S(h_p) = \begin{cases} S_r + \dfrac{S_{sat} - S_r}{[1 + (\alpha|h_p|)^n]^m} & \text{if } h_p < 0 \\ S_{sat} & \text{if } h_p \geq 0 \end{cases}
\tag{2}
$$

$$
K(h_p) = \begin{cases} K_s S_e^{0.5} \left[1 - \left(1 - S_e^{1/m}\right)^m\right]^2 & \text{if } h_p < 0 \\ K_s & \text{if } h_p \geq 0 \end{cases}
\tag{3}
$$

where

$$
S_e = \frac{S - S_r}{S_{sat} - S_r}
\tag{4}
$$

$[-]$ is the effective saturation and the model parameter $m$ $[-]$ is given here by $m = 1 - 1/n$. The model parameters $\alpha$ $[\text{L}^{-1}]$ and $n$ $[-]$ are related to the pore-size distribution, $S_{sat}$ and $S_r$ $[-]$ are the saturated and residual saturation, respectively.

### 2.1.2 Overland flow

If the water level rises above the land surface, the kinematic wave equation is solved to model overland flow:

$$
\frac{\partial h_s}{\partial t} = \nabla \cdot (\boldsymbol{v}(h_s)h_s) + q_r
\tag{5}
$$

where $h_s$ [L] is the surface ponding depth, $\boldsymbol{v}$ $[\text{LT}^{-1}]$ is the depth averaged two-dimensional velocity vector and $q_r$ $[\text{LT}^{-1}]$ is a source or sink term. The velocity is related to the ponding depth by Mannings equation:

$$
\boldsymbol{v}(h_s) = -\frac{\sqrt{S_f}}{n_f} h_s^{2/3}
\tag{6}
$$

with $S_f$ $[-]$ being the friction slope and $n_f$ $[\text{L}^{-1/3}\text{T}^{-1}]$ Manning's roughness coefficient. The two domains are coupled by adding a flux for subsurface exchanges $q_e$ $[\text{LT}^{-1}]$ to Eq. 5 and assuming that $h_p = h_s$ at the top soil cell. A more detailed description of the governing equations and their coupling can be found in Kollet and Maxwell (2006) and the ParFlow user manual (Maxwell et al., 2009). The equations are solved using an implicit Euler scheme for the discretization in time and a cell-centered finite difference scheme and an upwind finite volume scheme for the discretizations in space for Eqs. 1 and 5, respectively.





## 2.2 Ensemble Kalman filter

### 2.2.1 Analysis scheme

We use the Ensemble Kalman filter (EnKF) introduced by Evensen (1994) to handle the uncertainties associated with the hydrological model presented in the previous section. The EnKF uses an ensemble of model realizations to present the prior probability density functions (pdfs) of the uncertain model components. These uncertainties are then reduced by sequentially integrating information gained from (likewise uncertain) observations whenever these become available.

First, we need to define our model system which consists of the model states $x$, boundary $x_b$ and initial conditions $x_0$, time-invariant parameters $p$ and observations $y$. In our case, model states and observations are water content $\theta = S(h_p) \cdot \phi \, [-]$ in the entire domain and at the predefined observation locations, respectively. The model parameters are the chosen (sub)set of the soil hydraulic parameters $K_s$, $\phi$, $\alpha$ and $n$. We apply the augmented state vector approach (Evensen, 2009), i.e. we join all uncertain model quantities into one augmented state vector $\psi$. Since we assume our boundary conditions to be perfectly known, our augmented state vector consists of states and parameters: $\psi^T = [x^T, p^T]$. As mentioned above, the probabilistic representation of the uncertain states and parameters is achieved by using an ensemble of size $N$ where the realizations are samples of the state and parameter pdfs:

$$\Psi = \left( \psi^1, \psi^2, ..., \psi^N \right). \tag{7}$$

An underlying assumption of the EnKF is that all pdfs are Gaussian and can be fully described by its mean value $\mu$ and standard variation $\sigma$, which can be easily calculated from the ensemble. For the generation of the initial ensemble at the starting time $t_0$, the prior pdfs have to be prescribed, while the pdfs at later times are obtained by propagating the ensemble forward in time according to the forward model $\mathcal{F}$:

$$\Psi_k = \mathcal{F}(\Psi_{k-1}). \tag{8}$$

Here, $k$ denotes the time index and $\mathcal{F}$ is given by the equations in Sect. 2.1, and unity for the parameters.

States, parameters and observations are linked via the measurement operator $\mathcal{M}$:

$$\mathbf{Y}_k = \mathcal{M}(\Psi_k), \tag{9}$$

which in our case returns the water content (state) values at the observation locations and is thus a linear operator. These simulated observations are not to be mistaken for the real observations $y_k^{obs}$ obtained from measurements. However, these real observations are also uncertain as they are affected by measurement errors. To account for that, an ensemble of observations is generated by perturbing the measured observations with noise terms $\epsilon$ drawn from $\mathcal{N}(0, \varepsilon^2)$:

$$\mathbf{Y}_k^{obs} = \left( y_k^{obs} + \epsilon^1, y_k^{obs} + \epsilon^2, ..., y_k^{obs} + \epsilon^N \right) \tag{10}$$





where $\varepsilon$ is the measurement error.

The model and the observations are merged in the so-called *analysis step* which is performed at every time at which an
observation is available. An analysis step is preceded by a *forecast step* which integrates the augmented state ensemble from
the previous to the current observation time according to Eq. 8. During the analysis step the forecasted ensemble is updated to

$$\Psi_k^a = \Psi_k^f + \mathbf{K}_k \left( \mathbf{Y}_k^{obs} - \mathbf{Y}_k \right), \tag{11}$$

with $\Psi_k^f = \mathcal{F}(\Psi_{k-1}^a)$, $\mathbf{Y}_k^f = \mathcal{M}(\Psi_k^f)$ and $\mathbf{Y}_k^{obs} - \mathbf{Y}_k$ being the *innovation*, i.e. the difference between measured and simulated
observations. The superscripts $a$ and $f$ denote the ensemble after the forecast and after the analysis step, respectively. The
Kalman gain

$$\mathbf{K}_k = \mathbf{C}_{\psi y, k}^f \left( \mathbf{C}_{yy, k}^f + \mathbf{C}_{\epsilon\epsilon} \right)^{-1} \tag{12}$$

relates the covariances of states, parameters and measurements stored in $\mathbf{C}_{\psi y, k}^f$ to the covariances of the simulated observations
$\mathbf{C}_{yy, k}^f$ and the covariances of the measurement error $\mathbf{C}_{\epsilon\epsilon}$. The covariance matrices $\mathbf{C}_{\psi y, k}^f$ and $\mathbf{C}_{yy, k}^f$ can be easily calculated
from the ensemble as

$$\mathbf{C}_{\psi y, k}^f = \frac{1}{N-1} \sum_{i=1}^{N} \left( \boldsymbol{\psi}_k^{i,f} - \overline{\Psi}_k^f \right) \left( \boldsymbol{y}_k^{i,f} - \overline{\mathbf{Y}}_k^f \right)^T \tag{13}$$

and

$$\mathbf{C}_{yy, k}^f = \frac{1}{N-1} \sum_{i=1}^{N} \left( \boldsymbol{y}_k^{i,f} - \overline{\mathbf{Y}}_k^f \right) \left( \boldsymbol{y}_k^{i,f} - \overline{\mathbf{Y}}_k^f \right)^T. \tag{14}$$

with the overbar denoting the ensemble mean.

Equations 11 and 12 are obtained by maximizing the likelihood function of model states and parameters conditioned on the
given observations assuming Gaussian distributions. As shown in Evensen et al. (2009), this is equivalent to minimizing the
model uncertainty represented by the state and parameter variance $\mathbf{C}_{\psi\psi}$, defined analogously to Eqs. 13 and 14.

### 2.2.2   Technical specifications

The update of the forecasted ensemble in Eq. 11 adds the difference between measured and simulated observations, projected
onto the augmented state vector space by the state/observation covariance matrix, to the ensemble. This difference is further
scaled by the summed uncertainty of observed and measured observations. Thus, the increment is large

     – for states and parameters that show strong correlations to the observations





– for ensemble members whose simulated observations differ significantly from the measured value

– for the entire ensemble when the measurement error is small.

Due to the usage of an ensemble of finite size $N$, Eqs. 13 and 14 give only approximations of the real covariances which can cause spurious correlations. These may result in a wrong update of the ensemble which can lead to filter divergence on a longer term. Furthermore, at every analysis step, the spread of the ensemble is reduced as a consequence of the uncertainty minimization. Thus, the covariances may become underestimated allowing only for small increments even though the simulated observations may be far off the measured ones.

There exist different approaches to handle these issues. The problem of spurious correlations can be helped by either using a large enough ensemble or by applying localization. Localization reduces or eliminates entries in the covariance matrix where only small or no correlations are to be expected, e.g. when the spatial distance to the observation location is too large. As we did not encounter spurious correlations or a positive effect of localization, we assume the ensemble size in our experiments to be large enough so that localization is not needed.

However, we noticed a strong reduction of ensemble spread already during the first analysis steps, especially when performing parameter estimates. While the application of inflation, where the ensemble perturbations are artificially increased by a small factor (usually 1.01) after every analysis step, led to instabilities of our ensemble, we achieved better results by applying a *dampening factor* $\beta$ as described in Hendricks Franssen and Kinzelbach (2008):

$$\Psi_k^a = \Psi_k^f + \beta \mathbf{K}_k \left( \mathbf{Y}_k^{obs} - \mathbf{Y}_k \right), \tag{15}$$

with $\beta \in [0,1]$. In Equation 15, the dampening factor is applied directly to the augmented state ensemble, which in our case holds water content and the soil hydraulic parameters. The forward model (Eq. 1), however, depends strongly on the pressure head values $h_p$. Before each forecast step, the pressure heads need to be calculated out of the updated water contents by applying the inverse of Equation 2. Close to the residual water content $\theta_r \approx S_r \cdot \phi$, the pressure head is very sensitive to changes in water content. Thus, small updates of water content may result in large updates of pressure head that can cause numerical problems during the next forecast step. Because of that we apply the dampening directly to the pressure head values (comprised in the pressure head ensemble matrix $\mathbf{H}_p$) instead of the water content:

$$\mathbf{H}_p^a = \mathbf{H}_p^f + \beta \cdot S^{-1}(\Theta^a). \tag{16}$$

Note, that the inverse of Equation 2 ($S^{-1}$) is only defined for unsaturated conditions $\theta < S_s \cdot \phi$. This means that the update is restricted to unsaturated areas while saturated areas (e.g. groundwater) can only be affected by the updates indirectly during the subsequent model integration or by the parameter updates. Alternatively, one can use pressure head instead of water content as model state, as done for simpler one-dimensional setups e.g. in Erdal et al. (2014) and Brandhorst et al. (2017). However, this results in a nonlinear measurement operator and caused numerical problems in our more complex three-dimensional model.





Nevertheless, we had to deal with non-converging ensemble members during our simulations, i.e. ensemble members for which the numerical flow simulation to the next time step did not converge after the update. This issue is specific for soil models and has been seen very often in data assimilation with soils (Camporese et al., 2009; Rasmussen et al., 2015). The reason is

most probably that the updates create states that could hardly develop with the given parameters and boundary conditions. Such members were eliminated from the ensemble and replaced by converging members before the next analysis step to keep the ensemble size constant.

The Parallel Data Assimilation Framework PDAF (Nerger and Hiller, 2013), developed at the Alfred Wegener Institute, is used for the filter. The framework provides all required routines for the application of a large choice of data assimilation

methods, like the EnKF. It also includes generic interfaces that allow for a coupling to any numerical forward model. The coupling to ParFlow was implemented by Kurtz et al. (2016) within the TerrSysMP-PDAF modeling platform. The different data assimilation strategies were tested with a hillslope model, which is described below.

## 3 Setup

### 3.1 Numerical model

We investigate the effect of parameter updates on soil moisture estimates in a three-dimensional hydrologic hillslope model as shown in Fig. 1. It covers an area of $50\,\mathrm{m} \times 50\,\mathrm{m}$ and has a depth of $20\,\mathrm{m}$. The hillslope is rather steep with $S_{f,x} = 0.1$, causing lateral fluxes in the saturated as well as the unsaturated zone. At the bottom of the hillslope the water discharges into a small creek with $S_{f,y} = 0.01$ that directs the water towards the outlet. All lateral boundaries and the bottom boundary are closed so that water can leave the domain only via the outlet or through evaporation. A flux boundary condition corresponding to the

hourly varying net flux of precipitation and evaporation (P-E, Fig. 2) is imposed on the surface.

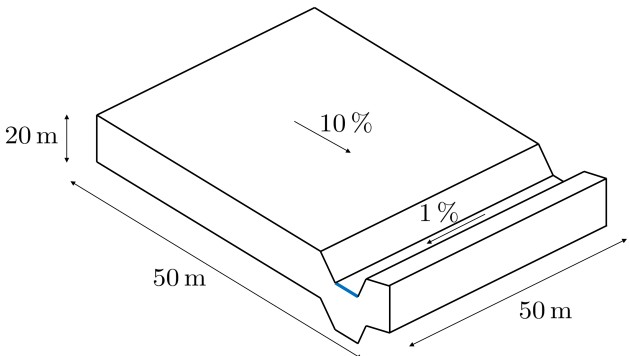

**Figure 1.** Dimensions and topography of the hillslope model. The blue line denotes the outlet.



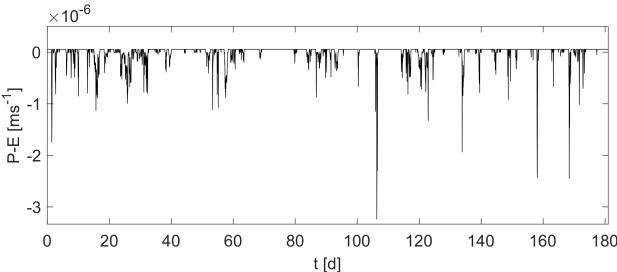

**Figure 2.** Net precipitation (negative) and evaporation (positive) time series.

The lower $18.6\,\text{m}$ of the subsurface consist of low permeable homogeneous bedrock while the upper $1.4\,\text{m}$ comprise either one or two loamy soil layers depending on the test case. The soil hydraulic parameter values are listed in Table 1. Manning's coefficient is $n_f = 4.2 \cdot 10^{-6}\,\text{m}^{-1/3}\text{h}$ for the entire land surface which has no vegetation.

The domain is discretized horizontally into a $50 \times 50$ grid with $\Delta x = \Delta y = 1\,\text{m}$. The vertical grid size is variable with
$\Delta z_{max} = 1.2\,\text{m}$ at the bottom and $\Delta z_{min} = 0.01\,\text{m}$ at the surface. The total number of cells in the vertical direction is $n_z = 50$. The total simulation time is 181 days, from the $1^{st}$ of January to the $30^{st}$ of June. The time step size is adaptive with a base value of $\Delta t = 0.1\,\text{h}$.

### 3.1.1 Reference parameter fields

The observations needed for the data assimilation are obtained from reference runs with the same numerical model and a
deterministic set of parameters. The usage of synthetic observations instead of field data allows to eliminate all unwanted sources of uncertainty, like model error, structural error, uncertain forcing etc., that most probably would strongly impact the state estimates. Furthermore, it offers full knowledge of all state and parameter values at each point in the domain and of the measurement error. To run the numerical reference model, we need to define a deterministic set of parameters which we deem the "true" parameters.

We set up two reference models for our experiments. The first model considers a homogeneous soil layer above the bedrock. Its values are given in Table 1. The homogeneous model is used to test whether the parameter updates work properly. For the main experiments a setup with two heterogeneous soil layers is chosen. The lower layer with a thickness of $z_{lower} = 1.31\,\text{m}$ exhibits a stronger heterogeneity while the heterogeneity of the upper layer ($z_{upper} = 0.09\,\text{m}$) is reduced to avoid numerical convergence issues. The mean values and standard deviations (in brackets) of the parameter fields are listed in Table 1. Note that
only the saturated hydraulic conductivity, porosity and the van Genuchten parameters are spatially distributed. This is because a previous sensitivity analysis showed negligible influence of the remaining parameters on the soil moisture which is in agreement with the findings of Brandhorst et al. (2017) and Bo et al. (2020) for one-dimensional unsaturated flow problems. Besides, $K_s$ and $\alpha$ follow a lognormal distribution as suggested by Carsel and Parrish (1988). The generation of the heterogeneous fields is constrained by the correlation coefficients of the parameters based on Carsel and Parrish (1988) and given in Table 2.





**Table 1.** Values of the soil hydraulic parameters for the different models. The values in the brackets are the standard deviations of the parameter distributions.

| | $K_s\,[\mathrm{mh^{-1}}]$ | $\phi\,[-]$ | $\alpha\,[\mathrm{m^{-1}}]$ | $n\,[-]$ | $S_r\,[-]$ | $S_{sat}\,[-]$ | $S_s\,[\mathrm{m^{-1}}]$ |
|---|---|---|---|---|---|---|---|
| | **all cases** | | | | | | |
| bedrock | $10^{-4}$ | 0.3 | 1 | 1.4 | 0.1 | 1 | $10^{-4}$ |
| | **homogeneous reference model** | | | | | | |
| soil | 0.02 | 0.42 | 3.5 | 2 | 0.1 | 1 | $10^{-4}$ |
| | **heterogeneous reference model** | | | | | | |
| upper soil | $0.015\,(0.11^{\mathrm{a}})$ | $0.39\,(5\cdot 10^{-4})$ | $4.26\,(0.05^{\mathrm{a}})$ | $2.25\,(0.005)$ | 0.1 | 1 | $10^{-4}$ |
| lower soil | $0.02\,(1.01^{\mathrm{a}})$ | $0.42\,(0.002)$ | $3.5\,(0.5^{\mathrm{a}})$ | $2\,(0.05)$ | 0.1 | 1 | $10^{-4}$ |
| | **homogeneous ensemble** | | | | | | |
| soil | $0.02\,(1^{\mathrm{a}})$ | $0.4\,(0.07)$ | $2.8\,(0.6^{\mathrm{a}})$ | $2\,(0.18)$ | 0.1 | 1 | $10^{-4}$ |
| | **heterogeneous ensemble**[b] | | | | | | |
| upper soil | $0.014\,(0.32^{\mathrm{a}})$ | $0.34\,(0.02)$ | $2.64\,(0.22^{\mathrm{a}})$ | $1.85\,(0.07)$ | 0.1 | 1 | $10^{-4}$ |
| lower soil | $0.008\,(0.71^{\mathrm{a}})$ | $0.45\,(0.03)$ | $3.67\,(0.34^{\mathrm{a}})$ | $1.8\,(0.1)$ | 0.1 | 1 | $10^{-4}$ |
| | **layered ensemble** | | | | | | |
| upper soil | $0.021\,(0.09^{\mathrm{a}})$ | $0.42\,(0.001)$ | $3.28\,(0.05^{\mathrm{a}})$ | $2\,(0.005)$ | 0.1 | 1 | $10^{-4}$ |
| lower soil | $0.021\,(1.12^{\mathrm{a}})$ | $0.4\,(0.01)$ | $8.44\,(0.5^{\mathrm{a}})$ | $2.04\,(0.05)$ | 0.1 | 1 | $10^{-4}$ |

a Assuming lognormal distributions.

b Statistics of individual parameter fields.

**Table 2.** Correlation coefficients of the soil hydraulic parameters.

| | $\log(K_s)$ | $\phi$ | $\log(\alpha)$ | $n$ |
|---|---|---|---|---|
| $\log(K_s)$ | 1 | $-0.4$ | 0.8 | 0.4 |
| $\phi$ | | 1 | $-0.2$ | $-0.6$ |
| $\log(\alpha)$ | | | 1 | 0.5 |
| $n$ | | | | 1 |

The initial conditions for both reference models are generated by spin-up runs by repeatedly applying the same forcing (Fig. 2) until a dynamic steady state is reached.





## 3.2 EnKF

An ensemble consisting of 100 members is used for the data assimilation. Each ensemble member is an identical copy of the reference run with only the soil hydraulic parameters $K_s, \phi, \alpha$ and $n$ of the soil layer(s) being changed. The other parameters

were found to be non-sensitive and are therefore assumed to be perfectly known. Since the initial condition is generated by a spin-up run, the different parameter sets lead to different initial conditions of the model states for the individual ensemble members. Thus, in addition to the parameter uncertainty, there is a model error caused by an uncertain initial condition.

The observations are obtained from the reference model runs described in the previous section. We use observations of soil moisture which are taken hourly at four measurement locations as shown in Fig. 3. Two locations are situated downhill (1 and

2) and the other uphill (3 and 4). Additionally, the soil moisture is measured at two validation locations in the center of the hillslope (5) and at its upper corner (6). These measurements are only used for the evaluation of the filter performance and not for the analysis step in the filter. At each of the six locations, the soil moisture is taken at three depths, $z_1 = 0.33\,\text{m}$, $z_2 = 0.75\,\text{m}$ and $z_3 = 0.95\,\text{m}$, below the surface, summing up to a total of twelve measurement locations and six validation locations. Even though hourly observations are available, the updates are performed only daily to increase the computational efficiency and

because it has been shown that higher assimilation frequencies have a negative impact on the soil moisture estimates (Valdes-Abellan et al., 2019). The soil moisture values from the reference runs are perturbed with random white noise with a standard deviation of $\varepsilon = 0.01$ to account for the measurement error. In the results section, we will only show soil moistures for one observation borehole (#1) and one validation borehole (#5) to avoid too many plots. The findings would not change if the other boreholes would be shown. It should, however, be noted that the values from all boreholes are included in the error calculations.

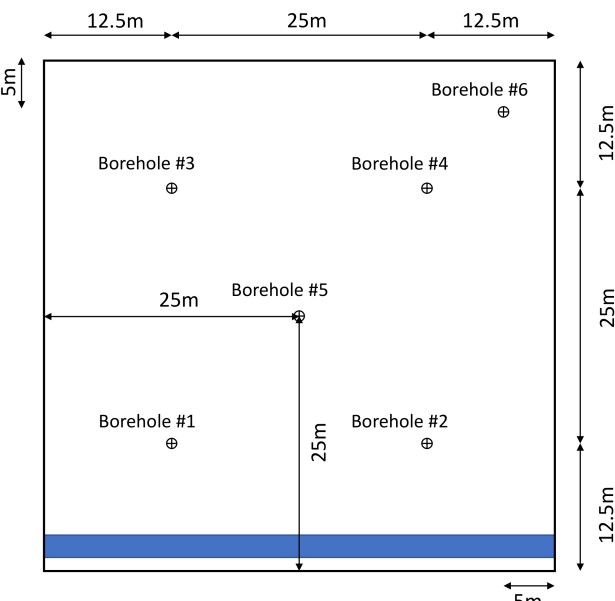

**Figure 3.** Observation (1-4) and validation (5 and 6) locations. The creek is plotted in blue.



The analysis step is performed according to Eqs. 15 and 16. A preliminary investigation suggested to use a dampening factor $\beta = 0.1$ for states and parameters. Non-convergent ensemble members are eliminated from the ensemble and replaced by randomly chosen converging members. Note that we replace only the values contained in the augmented state vector while all other parameters remain at their original values to maintain the ensemble diversity.

### 3.2.1    Parameter ensembles

We generate three ensembles that differ in terms of their soil setup. The first one considers one homogeneous soil layer similar to the homogeneous reference model. The ensemble mean and standard deviation (in brackets) for each parameter are listed in Table 1. The initial guesses of $K_s$ and $n$ correspond almost to the true values while the other two have a bias ($\phi$ and $\alpha$). This allows to investigate how the parameter ensembles behave when the initial guess is "correct" and when it is "wrong". The large ensemble spreads account for a large initial parameter uncertainty and shall counteract a too fast decrease of the ensemble

spread.

    The second ensemble is based on the heterogeneous reference model with two heterogeneous soil layers. Each parameter field is generated with a random field generator constrained by the mean and standard deviation (in brackets) given in Table 1 and the correlations given in Table 2. One has to keep in mind that the statistics of the individual parameter fields are not identical with those of the estimated parameter field (ensemble mean). While the correlation coefficients and the mean value are very similar, the standard deviation is much smaller. Thus, the initial guesses of the soil layers are almost homogeneous.

Besides, there is a large bias in the initial parameter guesses at the grid points. It should be noted that using this ensemble, the EnKF implies that all parameters in the field (no. of grid nodes × no. of parameters) are updated in the augmented state vector. We suspect that this will not lead to good estimates of the parameter fields, but are interested in its impact on the soil moisture estimates.

In the third ensemble again two soil layers are considered. While the depth of the layers is the same as in the heterogeneous reference model, the soil layers are homogeneous in this case. The parameter values are sampled from Gaussian distributions with mean and standard deviation (in brackets) as given in Table 1. The values are chosen to create a biased initial ensemble with a spread that is large enough to give a realisitic estimate of the parameter uncertainty and small enough to prevent major numerical problems.

We assume the depth of the layers to be known. This is a reasonable assumption since this information can be obtained from a borehole sample and is not expected to vary significantly within such a small domain. Nevertheless, we are aware that this can be a relevant source of uncertainty, especially for large scale models. Yet, this is not part of this work and we refer to existing studies on this subject as Erdal et al. (2014).

### 3.3    Test series

We perform three test series with the described reference models and ensembles. The first test series involves the homogeneous reference model and the homogeneous ensemble. It serves as a testbed for the implemented parameter updates and analyzes





the impact of parameter updates under two-dimensional flow conditions in the absence of soil heterogeneities. Even though the model is three-dimensional, the topography and the homogeneity of the subsurface cause a quasi two-dimensional flow field.

Real soils are certainly not homogeneous but heterogeneous. As outlined in the introductory part of this work, this leads to
model uncertainty. We want to investigate how to deal with this uncertainty when the goal is to get decent predictions of soil moisture. Hence, for the other test series, the heterogeneous model is used as reference. Often, the heterogeneity is neglected. We compare this approach to resolving the heterogeneity. This will most probably lead to wrong estimates of the parameter fields, but fields with an "equally" heterogeneous structure. Therefore, in the second test series the heterogeneous ensemble is used. With these experiments, on the one hand, we want to investigate how different parameter updating strategies could
improve soil moisture estimates in a three-dimensional heterogeneous model which would be the case for any field application. On the other hand, we want to compare the estimates to those obtained when the heterogeneous structure is neglected. These are the results of the third test series where we use a simplified (layered) soil structure in the ensemble and try to represent the soil moisture of the heterogeneous reference model. Using a layered structure with homogeneous layers would significantly reduce the size of the augmented state vector but may lead to less accurate state estimates.

As we want to identify the parameters which lead to the best soil moisture estimates when included in the update, we perform joint updates of soil moisture $\theta$ and all possible combinations of the sensitive soil hydraulic parameters, $K_S, \phi, \alpha$ and $n$. The results are compared to those of an open loop (OL) run, where the ensemble is propagated forward in time without any updates, and a run where only $\theta$ is updated. To reduce the total number of model runs, we treat the van Genuchten model parameters $\alpha$ and $n$ as a unit and update either both or none of these two.

It should also be noted that we consider only "well behaving" heterogeneity, so multi-Gaussian fields with correlation lengths much smaller than the domain size. We are aware that heterogeneous structures in real soils are much more complex (Schlüter et al., 2012) but we are confident that our findings are general for the applied method and not bound to the specific soil structure of this test case.

## 4 Results and Discussion

The results of the data assimilation runs are compared with regard to the soil moisture estimates by means of the spatially and temporally averaged *root mean square error $RMSE\,[-]$* at the observation and validation locations, respectively:

$$RMSE = \frac{1}{n_t} \sum_{k=1}^{n_t} \sqrt{\frac{\sum_{i=1}^{n_i} \left( \overline{\theta}_{k,i} - \theta_{k,i}^{ref} \right)^2}{n_i}} \qquad (17)$$

with $n_t = 362$ being the number of output time steps (twice a day, not to be mistaken with the assimilation frequency which is once a day), $n_i$ the number of grid cells at the observation (12) or validation (6) locations, respectively, $\overline{\theta}$ the soil moisture
estimate (ensemble mean) and $\theta^{ref}$ the soil moisture in the reference model. Furthermore, we compare the number of converging ensemble members as a criterion for the numerical stability. As was already mentioned in the introductory section, the updates can lead to unphysical parameter-state combinations and thus to convergence problems and long run times. Therefore,





the numerical stability of the ensemble is an important factor in the evaluation. One has to keep in mind that the non-converging

members are replaced during the assimilation. Hence, we only consider the non-converging members in the time after the last

update when the ensemble has reached its final state.

We also look into the parameter estimates. These can help understand the performance of the filter with regard to the soil

moisture estimation as they can be e.g. a reason for filter divergence. Note that the states are always included in the update

even though we will not always explicitly mention this.

### 4.1    Results of the homogeneous test case

Figure 4 shows the soil moisture over time at two boreholes (one observation location and one validation location) and $z_1 =$

$0.33\,\mathrm{m}$ below the surface, exemplarily for the data assimilation runs without parameter updates ((a) and (b)) and when updating

all parameters ((c) and (d)). Loose ends of the ensemble members (gray lines) are caused by the resampling of non-converging

realizations as explained in Sect. 2.2.2. It can be seen that in both cases the data assimilation improves the soil moisture

estimates at the observation location compared to those of the open loop run. Indeed, the reference soil moisture is perfectly

matched after some time. This happens faster for the joint update, converging to the reference soil moisture after $\approx 40$ days

compared to $\approx 80$ days without parameter updates. The parameter updates cause a significant decrease of the ensemble spread.

The spread can be expressed in terms of the standard deviation of the ensemble $\sigma_e$. While $\sigma_e = 0.027$ without parameter

updates, it is only $\sigma_e = 0.001$ for the joint update of all parameters. These values refer to the soil moisture at the last time step

at the shown observation location. This means that in this case in Eq. 11 the ensemble estimates are deemed more certain than

the measurements and correlations are very small impeding further updates.

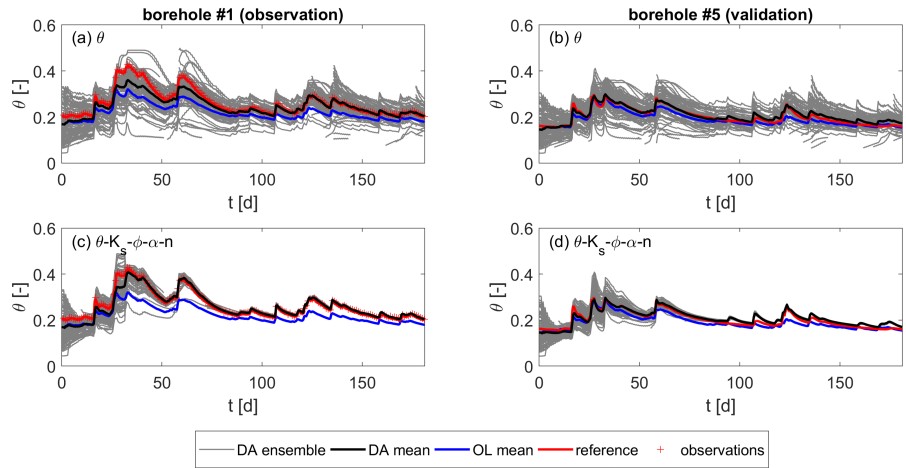

**Figure 4.** Soil moisture over time for the homogeneous scenario. Values are taken at one observation and one validation location, each
at $z_1 = 0.33\,\mathrm{m}$ below the surface. Plots (a) and (b): updates of soil moisture only; plots (c) and (d): joint updates of soil moisture and all
parameters.





At the validation location (Fig. 4 (b) and (d)) the estimates are improved in the first half of the time series but towards the end of the simulation the open loop estimates are closer to the reference soil moisture, albeit the difference is small. Again, the strong reduction of the ensemble spread for the joint update is obvious. This can be the reason why the estimates towards the end of the time series are worse. As the estimates of the run without parameter updates are of comparable accuracy even

though the ensemble spread is still large, it seems more likely that there is another reason for the deviations. This could be the small difference between estimated and measured soil moisture at the observation locations which is an important factor in the update equation (Eq. 11).

To evaluate the overall performance of the different parameter updates, the mean $RMSE$ values as given by Eq. 17 are plotted in Fig. 5 for the observation and validation locations, respectively. The measurement error and the $RMSE$ of the open

loop run are plotted in the dotted and dashed line, respectively, for comparison. The large markers denote the combination that led to the best result. Here, this is the combination of jointly updating soil moisture $\theta$, porosity $\phi$ and the van Genuchten model parameters $\alpha$ and $n$. For this run, the $RMSE$ is close to the measurement error at all considered locations. The second best result was achieved by including all parameters in the updates. Generally, it can be seen that the data assimilation improves the soil moisture estimates compared to the open loop run (even though we have seen that at some locations and time steps it can

be the opposite case) and that the parameter updates improve the estimates compared to updating only states.

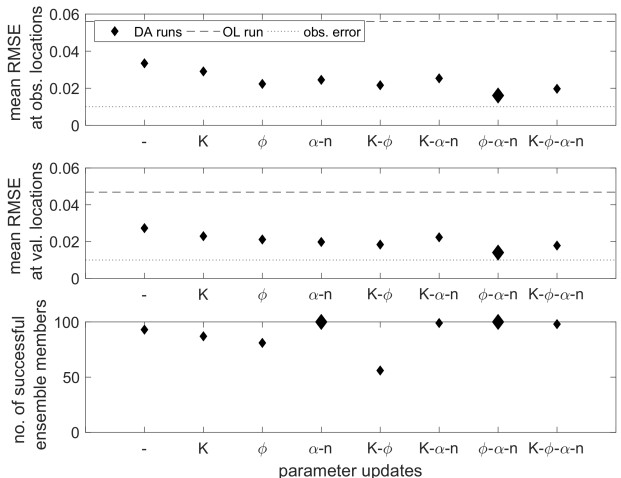

**Figure 5.** Spatially and temporally averaged $RMSE$ values at the observation and validation locations for the homogeneous scenario and the different parameter update combinations. The first entry (-) denotes the data assimilation run where only soil moisture is updated. The lower plot gives the number of converging ensemble members. The best values are highlighted by larger markers.

The lower plot of Fig. 5 gives the number of converging ensemble members. There are two runs where the entire ensemble is converging, namely when updating the van Genuchten parameters with or without porosity. Besides, for the combination $K_s$-$\alpha$-$n$ we get 99 converging members and for $K_s$-$\phi$-$\alpha$-$n$ 98 members. This leads to the conclusion that updating the van





Genuchten model parameters is crucial for the numerical stability of the ensemble. On the other hand, when only the saturated
hydraulic conductivity and porosity are updated, almost half of the ensemble fails.

The probability density functions (pdfs) of the soil hydraulic parameters are shown in Fig. 6. The dashed line indicates the
initial ensemble while the solid line represents the pdf at the end of the assimilation run updating all parameters. The reference
value is plotted in red. The initial pdfs have a large spread and a small bias. The final pdfs have a very small spread which is
consistent with the small ensemble spread for the soil moisture. Except for $n$, the parameter estimates of all other parameters
are closer to the reference value than their initial guesses. One has to keep in mind that the small deviations in the lognormal
distributions of $K_s$ and $\alpha$ correspond to larger deviations in the not log-transformed ensemble. For $n$ the final guess is slightly
worse even though the initial guess was almost equal to the reference value.

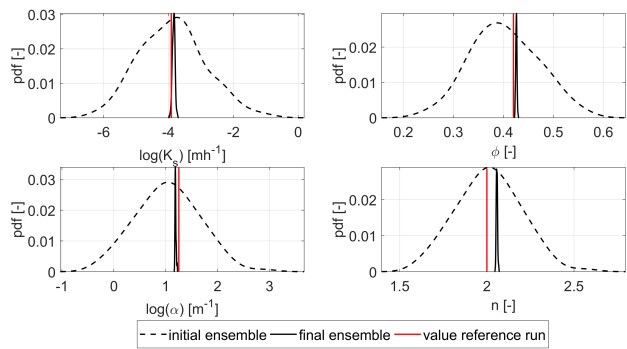

**Figure 6.** Probability density functions (pdfs) of the parameters for the homogeneous scenario and the joint update of all parameters.

From this simple test scenario we can draw the following conclusions:

1. The implemented parameter updates and the data assimilation work properly.

2. Even in such a simple scenario with little biased initial guesses the parameters do not fully converge to their true values.

3. Small parameter errors already lead to visible deviations of the soil moisture estimates.

The latter point can be seen from Fig. 4 (d). Reference and estimated soil moisture coincide well during the first part of the
time series and start to deviate in the second half. Due to the small ensemble spreads, which only allow for small changes
during the analysis step, this deviation cannot be caused by the filter updates but has to be due to the parameters differing
slightly from their reference values. This effect can also be seen at other locations which are not shown here. Conclusions on
the performance of the different parameter updates are discussed in Sect. 4.4 taking into account the results of all three test
series.

## 4.2  Results of the heterogeneous test case

The heterogeneous test case is discussed next. With this we want to test the transferability of the findings in simple, homoge-
neous settings to more realistic, heterogeneous ones. The soil moisture over time of this test case is plotted for one observation





and one validation location in Figs. 7 and 8, respectively. The plots on the left correspond to the run with state updates only, while the plots on the right are obtained by jointly updating all parameters. The values are taken at the three measurement depths, $z_1 = 0.33\,\mathrm{m}$ ((a) and (b)), $z_2 = 0.75\,\mathrm{m}$ ((c) and (d)) and $z_3 = 0.95\,\mathrm{m}$ ((e) and (f)) below the surface. Again, the data assimilation positively influences the soil moisture estimates compared to those of the open loop run. However, if the soil is

saturated, the state updates have no effect as the update is limited to the unsaturated zone (see Sect. 2.2.2). An indirect influence of the updated unsaturated parts on the saturated zone during the model forecast cannot be seen. At locations, that are - temporarily - below the groundwater level (Fig. 7 (c) and (e)), the open loop and data assimilation estimates coincide. On the other hand, when the parameters are updated, especially the porosity since $\phi = \theta_{sat}$ in our case, the estimates match the observations even when the soil is saturated. Under unsaturated conditions (Fig. 7 (a) and (b)), the estimates for the joint update

are more accurate, too, although the difference is smaller. Just like for the homogeneous test case, the parameter updates cause a significant decrease of the ensemble spread at the observation locations which can be either good or problematic as will be discussed in Sect. 4.4.

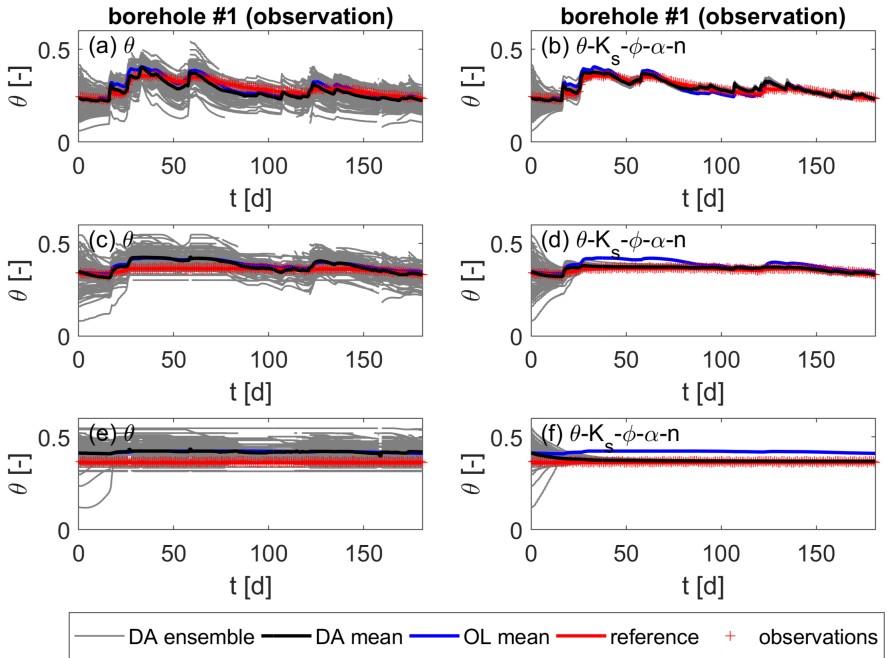

**Figure 7.** Soil moisture over time for the heterogeneous scenario at one observation location. Values are taken at $z_1 = 0.33\,\mathrm{m}$, $z_2 = 0.75\,\mathrm{m}$ and $z_3 = 0.95\,\mathrm{m}$ below the surface from top to bottom. Plots (a), (c) and (e): updates of soil moisture only; plots (b), (d) and (f): joint updates of soil moisture and all parameters.

At the validation locations (Fig. 8), though, the ensemble spread is hardly reduced for both assimilation runs. The parameter spread at the end of the simulation time is still large at the validation locations (see Fig. 9) which causes the large spread in

soil moisture. This means that the correlations between parameters and states at the validation locations and the observations





are too small to induce an update of the former. Thus, the parameter and state uncertainty at those locations is still large after the assimilation. Actually, the reference soil moisture is predominantly not enclosed by the ensemble indicating that the ensemble spread is still too small to correctly represent the model error. Nevertheless, there is an improvement by the data assimilation compared to the open loop run and the joint update of all parameters (right plots of Fig. 8) further clearly improves

the estimates. This is, however, too small to draw conclusions about the filter strategy.

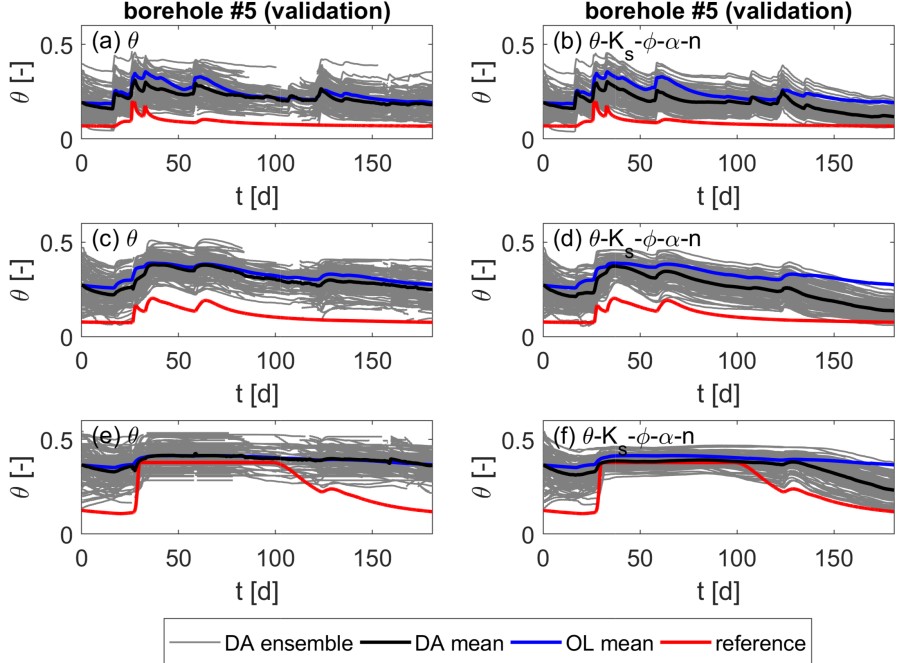

**Figure 8.** Soil moisture over time for the heterogeneous scenario at one validation location. Values are taken at $z_1 = 0.33\,\mathrm{m}$, $z_2 = 0.75\,\mathrm{m}$ and $z_3 = 0.95\,\mathrm{m}$ below the surface from top to bottom. Plots (a), (c) and (e): updates of soil moisture only; plots (b), (d) and (f): joint updates of soil moisture and all parameters.

Comparing all parameter update combinations based on the averaged $RMSE$ values (Fig. 10) the difference in accuracy at the observation and validation locations becomes evident. The $RMSE$ is roughly four times higher at the latter for all runs. This illustrates the strong influence of the soil heterogeneity on local estimates. Similar to the first test series, the data assimilation improves the estimates albeit the improvement is small at the validation locations. Parameter updates further

reduce the $RMSE$ at all locations, except for when updating $K_s$ and the van Genuchten model parameters. Also, for this combination the number of non-converging ensemble members is 22 and thus the highest for this test series. This assimilation run is a good example for the failure of unsaturated zone parameter updates. The updates cause non-physical parameter-state combinations which lead to numerical problems and eventually to worse estimates. The best results regarding the estimates at the observation locations and the numerical stability is obtained when updating all four parameters. Yet, the differences among

the parameter update combinations are small.





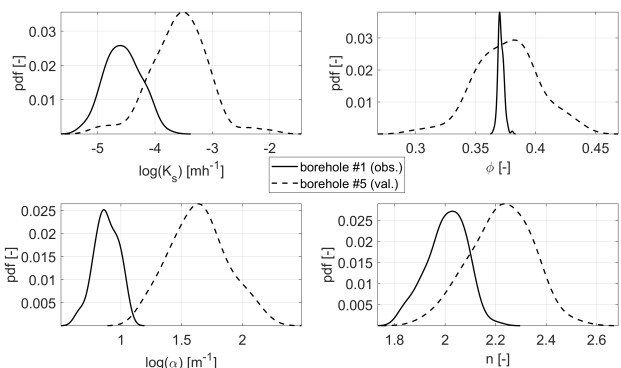

**Figure 9.** Estimated probability density functions (pdfs) of the four parameters for the heterogeneous scenario at one observation and one validation location and $z_2 = 0.75$ m below the surface after the last update.

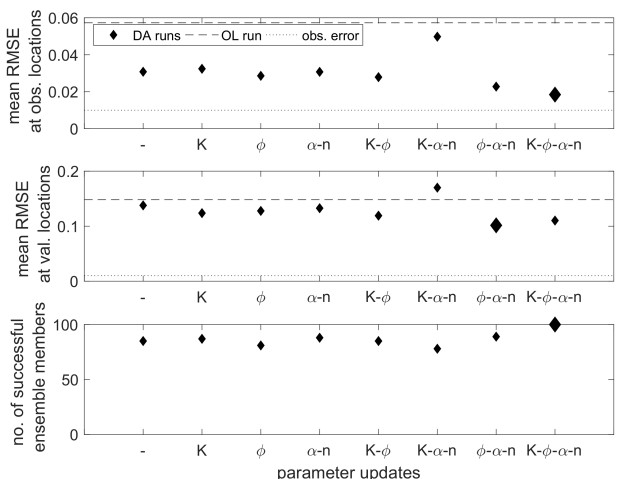

**Figure 10.** Spatially and temporally averaged $RMSE$ values at the observation and validation locations for the heterogeneous scenario and the different parameter update combinations. The first entry (-) denotes the data assimilation run where only soil moisture is updated. The lower plot gives the number of converging ensemble members. The best values are highlighted by larger markers.

Figure 11 shows the *normalized mean deviation $NMD$ [−]* of the estimated parameter fields for the two soil layers. The estimated parameter fields are calculated as the ensemble mean. The $NMD$ is a measure of the discrepancy between reference and estimated fields at each grid point, that is at the "point scale":

$$NMD = \frac{\sum_{i=1}^{n_i} |\overline{p}_i - p_i^{ref}|}{|\sum_{i=1}^{n_i} p_i^{ref}|}.$$  (18)

Here, $n_i$ is the number of grid points in the current soil layer, $\overline{p}$ is the ensemble mean parameter value and $p^{ref}$ the true parameter value used for the reference model run. Note that the initial guess of the parameter fields does not change in the





open loop run and when updating only soil moisture. The error bars in Fig. 11 illustrate the standard deviation of the $NMD$. Only the data assimilation run updating all four parameters is considered in the following analysis which allows to look into the estimated fields of all uncertain parameters. An analysis of the parameter estimates for all update combinations would be

confusing and is out of the scope of this work where the focus is on the soil moisture estimates. In Fig. 11 it becomes evident that the estimated parameter fields are not improved by the parameter updates at the point scale. Mostly, the $NMD$ is equal or higher to that of the initial guess. The standard deviation of the $NMD$ makes clear that there can be large deviations at some locations, especially for the van Genuchten parameter $\alpha$ in the lower soil layer. This confirms the hypothesis made in Sect. 3.2.1 that it is not possible to retrieve the heterogeneous reference parameter field by the assimiliation.

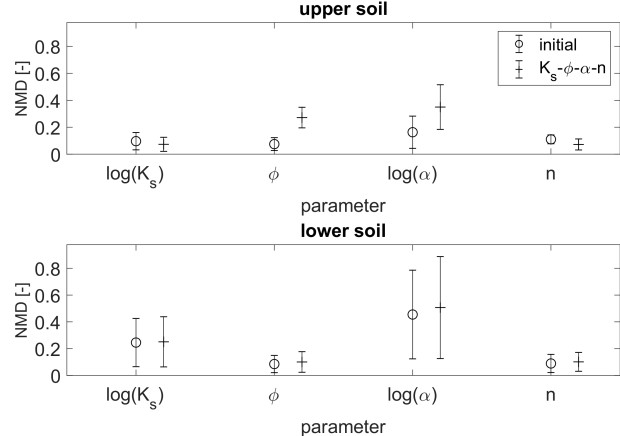

**Figure 11.** Normalized mean deviation $NMD$ of the estimated parameter fields compared to the reference field for the two soil layers of the heterogeneous scenario. The error bars denote the standard deviation.

Nevertheless, the updates are able to recover some features of the reference parameter field which can be seen in Figs. 12 and 13. The two figures show a horizontal and a vertical cut through the reference, initial and estimated $K_s$ field, respectively. While the initial guess of the field is rather smooth, caused by the averaging of the initial ensemble realizations, the structure of the estimated $K_s$ field is clearly more similar to that of the reference field. The regions of high or low values do not necessarily coincide with the reference, but the frequency distribution of the parameter values is similar. Furthermore, the layered structure

of the soil is maintained during the updates (see Fig. 13). We see the same behavior for the fields of the other three parameters.

     Hence, we now compare the reference and the estimated parameter fields in terms of their statistics. To this purpose we calculate the normalized mean value $NMV$ [-]

$$NMV = \frac{\sum_{i=1}^{n_i} \overline{p}_i}{\sum_{i=1}^{n_i} p_i^{ref}}. \tag{19}$$

     and its standard deviation indicating the spatial variability of the field. For the reference field, the $NMV$ is therefore equal

to one. In contrast to the $NMD$ shown in Fig. 11, this is a comparison at the "field scale". Figure 14 shows the $NMV$ and the





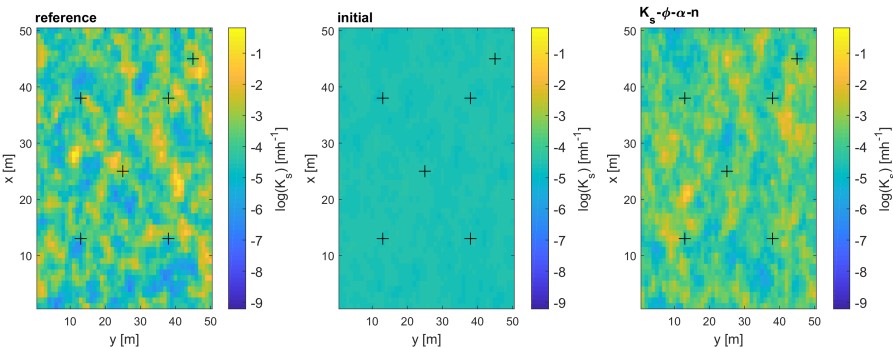

**Figure 12.** Horizontal cut of the saturated hydraulic conductivity field at $z_2 = 0.75$ m below the surface for the reference run, the initial ensemble and the estimated field when updating all parameters. The crosses denote the locations of the observation and validation boreholes. Note that the middle and the right field are ensemble means.

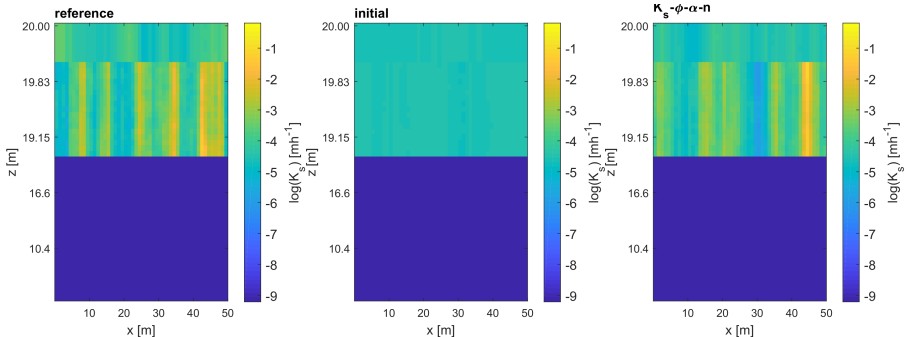

**Figure 13.** Vertical cut of the saturated hydraulic conductivity field at $y = 25$ m for the reference run, the initial ensemble and the estimated field when updating all parameters. The $z$-axis is scaled for better visibility. Note that the middle and the right field are ensemble means.

standard deviation of the four parameter fields for the two soil layers, respectively. While the mean values are generally not improved by the update, the standard deviation, which is systematically too small in the initial guess, is increased significantly for all parameters approximating its reference value. Regarding the statistics, we get a decent estimate of the saturated hydraulic conductivities, while the estimates of the van Genuchten $\alpha$, on the contrary, are improved but still rather poor.

Such a large uncertainty of the parameter estimates should be represented by a large ensemble spread. The ensemble spread can be illustrated by the cumulative density functions (cdfs) of all parameter values in the ensemble which are shown in Fig. 15 for each of the two soil layers. For the ensembles, two lines are plotted, one indicating the minimum values and the other the maximum values. The area between those lines thus represents the parameter values contained in the ensemble. The cdf of the reference field is plotted with a single solid line for comparison. For all parameters and layers we see a reduction of the
ensemble spread compared to the initial guess. However, compared to the posterior pdfs of the homogeneous test scenario (Fig. 6), the spreads are clearly higher in this case. The parameter values of the reference run are mostly enclosed by the final





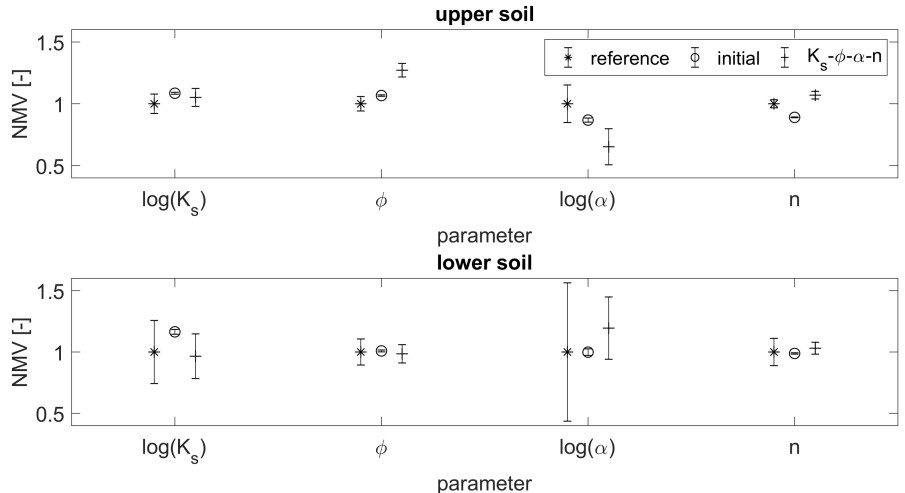

**Figure 14.** Statistics (mean and standard deviation) of the parameter fields of the two heterogeneous soil layers. All values are normalized by the mean value of the reference field. The statisctics of the ensemble runs refer to the estimated parameter field (ensemble mean).

ensemble. One exception is the porosity of the upper soil layer where we have already seen in Fig. 14 that the estimated mean value is too high. Yet, for the van Genuchten parameter $\alpha$, where the estimates have a large bias, too, the ensemble spread is still just large enough to contain the reference values. For the lower soil layer we even see an approximation towards the extreme values compared to the initial ensemble, although these remain outside of the ensemble spread.

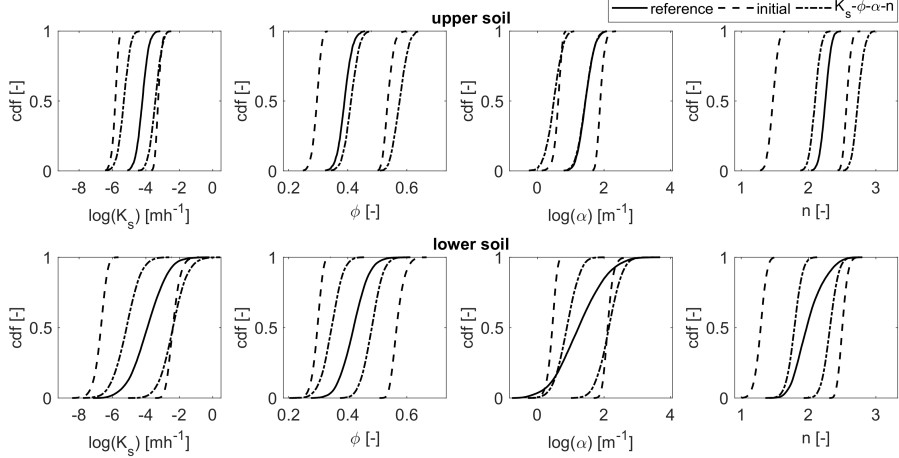

**Figure 15.** Cumulative density functions (cdfs) of the parameter values for the two heterogeneous soil layers. The solid line denotes the cdf of the reference parameter field. For the initial (dashed) and estimated (dash-dotted) ensemble, two lines are plotted indicating the minimum and the maximum values of the ensemble.





In addition to the conclusions of the test series with the homogeneous scenario, we can now further summarize:

1. Small correlations of the observations to parameters at other locations cause only small updates of the latter maintaining a high uncertainty of the soil moisture estimates there. Soil moisture estimates at observation locations are improved but not at points away from the observations.

2. Consequently, in presence of soil heterogeneities soil moisture estimates are less accurate aside the observation locations.

3. Generally, the parameter estimates are better in the lower soil layer that contains the observations. This does not refer to point values but to the field statistics.

4. Point values of the parameter fields differ clearly from the values of the reference field.

5. The representative variability of the parameter fields is improved by the data assimilation updating the soil parameters.

**4.3 Results applying a simplified soil structure**

In the third test series the soil moisture of the heterogeneous reference run is represented by an ensemble that consists of two homogeneous soil layers. The reason behind this is that, as shown before, heterogeneous fields cannot be retrieved, so the missing information shall not be included in the model. Based on the results of the heterogeneous scenario, for this test series, only two data assimilation runs were performed, one without parameter updates and one updating all four parameters. Figures

16 and 17 show the corresponding soil moisture over time at one observation and one validation location, respectively. The values are taken at three depths, $z_1 = 0.33\,\mathrm{m}$, $z_2 = 0.75\,\mathrm{m}$ and $z_3 = 0.95\,\mathrm{m}$ below the surface. In Fig. 16 the positive influence of the data assimilation can be seen. The estimated soil moisture is clearly closer to the reference soil moisture as in the open loop run. Nevertheless, there remains a deviation from the reference soil moisture. Furthermore, this deviation is not reduced by the parameter updates (Fig. 16 (b), (d) and (f)).

At the validation location shown in Fig. 17 we see a failure of the data assimilation. The estimated soil moisture is further away from the reference soil moisture than when performing an open loop run without updates. Again, there is no improvement by updating the soil hydraulic parameters. In contrast to the results of the heterogeneous ensemble (Fig. 8) the ensemble spread is reduced significantly by the parameter updates. Due to the homogeneity of the soil layers the parameters at the validation locations and the observations are fully correlated and the parameter ensemble is updated within the entire soil. Since the

parameters at the observation and validation locations differ in the reference model, this means that the updated values at the validation locations are a bad representation of the reference values leading to poor soil moisture estimates.

As a consequence the estimated soil moisture when updating the parameters is even less accurate than when updating only states which becomes evident in Fig. 18. The averaged $RMSE$ values at the observation and validation locations are plotted in Fig. 18 along with the corresponding values when using a heterogeneous ensemble (second test series). At the observation

locations the data assimilation is successful, reducing the $RMSE$ compared to the open loop run, even though the estimates are not as accurate as when applying a heterogeneous ensemble. At the validation locations the soil moisture estimates are comparable when updating only soil moisture and clearly better for the heterogeneous ensemble when including parameter





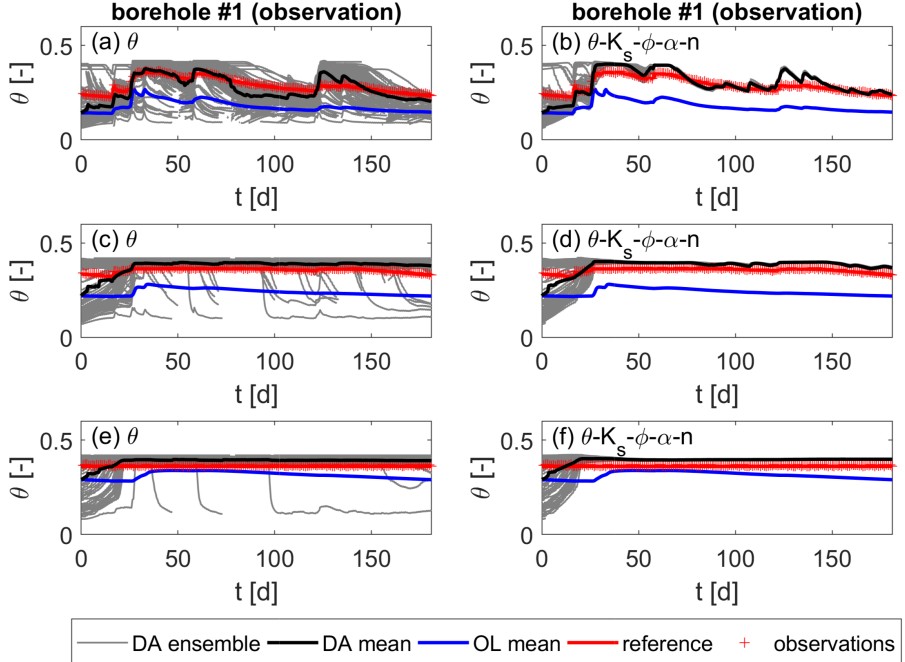

**Figure 16.** Soil moisture over time for the heterogeneous scenario with a simplified layered soil in the ensemble runs at one observation location. Values are taken at $z_1 = 0.33\,\mathrm{m}$, $z_2 = 0.75\,\mathrm{m}$ and $z_3 = 0.95\,\mathrm{m}$ below the surface from top to bottom. Plots (a), (c) and (e): updates of soil moisture only; plots (b), (d) and (f): joint updates of soil moisture and all parameters.

updates, despite the fact that the open loop $RMSE$ is much smaller for the layered ensemble. The wrong correlations generated by the homogeneous soil layers lead to wrong updates and thus worse soil moisture estimates.

On the other hand, the layered ensemble is numerically more stable, with 97 and 100 converging ensemble members for the two data assimilation runs, respectively (see lower plot of Fig. 18).

The third plot from the top in Fig. 18 shows the spatially averaged RMSE values for the root zone soil moisture. The root zone soil moisture is here defined as the spatially averaged soil moisture in the lower soil layer. It is an important quantity regarding the water supply for potential plants growing on the hillslope. Here, it shall be used as a criterion to evaluate the

ability of the different soil structures in the ensemble to represent the mean behavior of the reference run in contrast to the point values analyzed above. Figure 19 shows the results for the data assimilation runs using the heterogeneous ensemble and the runs using the layered ensemble, both when updating only states or states and all four parameters, respectively. First of all, it can be seen that, here, the data assimilation is successfull for all cases leading to better results than the corresponding open loop runs, especially for the layered ensemble where the open loop estimates are pretty poor. Besides, the $RMSE$ values are further

reduced by the parameter updates in which the improvement is more distinct for the layered soil structure. When updating all parameters (plots (c) and (d)), the accuracy of both runs are comparable. However, the spread in the layered ensemble is

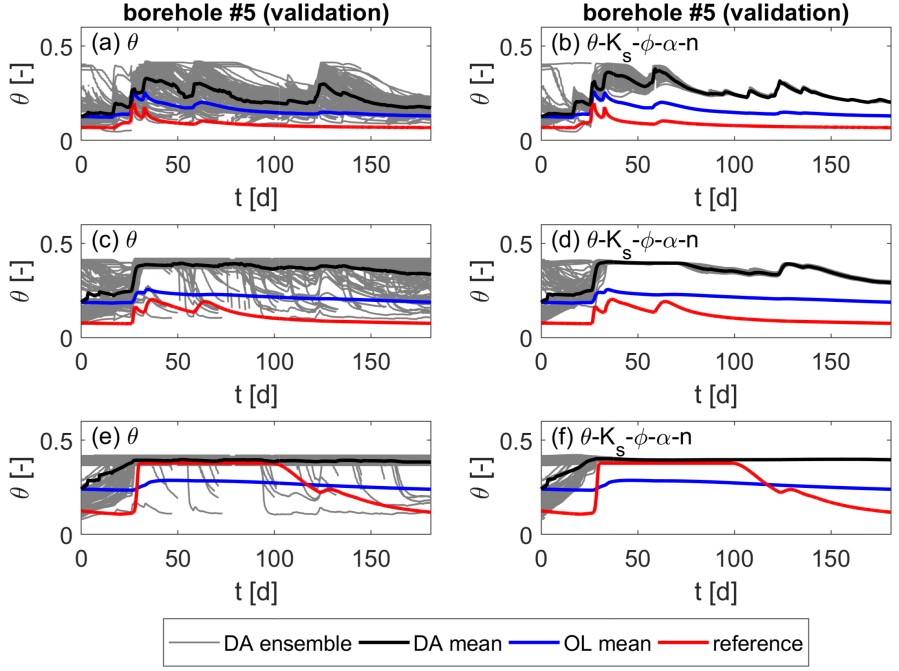

**Figure 17.** Soil moisture over time for the heterogeneous scenario with a simplified layered soil in the ensemble runs at one validation location. Values are taken at $z_1 = 0.33\,\mathrm{m}$, $z_2 = 0.75\,\mathrm{m}$ and $z_3 = 0.95\,\mathrm{m}$ below the surface from top to bottom. Plots (a), (c) and (e): updates of soil moisture only; plots (b), (d) and (f): joint updates of soil moisture and all parameters.

much smaller than in the heterogeneous ensemble and does not comprise the true state at the end of the simulation. The model uncertainty is thus underestimated in this case.

The (only temporally) averaged $RMSE$ values for the root zone soil moisture in Fig. 18 show that the accuracy of the
estimates of the layered ensemble approaches those of the heterogeneous ensemble by the updates: While the $RMSE$ of the layered ensemble is much higher than that of the heterogeneous ensemble for the open loop run, the difference is already smaller when performing state updates and gets very small when including parameter updates.

### 4.4 Discussion

The experiments demonstrate the strong influence of the soil hydraulic parameters on the soil moisture estimates. The ensemble
spread of soil moisture depends mainly on the parameter spread and cannot be reduced by state updates only. Thus, if there is a large uncertainty in the parameters, the estimated soil moisture will be uncertain as well. The aim of the data assimilation is to reduce the uncertainties in the model. However, the updates can cause a too strong reduction of the ensemble spread which means that the actual uncertainty is underestimated. If the estimates match the true states, this is not a problem since the model uncertainty is truly very small. This is e.g. the case in the homogeneous test case when performing parameter updates (Fig. 4
(c)). Otherwise, the small spread can lead to filter divergence as can be seen in Fig. 17 (b), (d) and (f). This is unfavorable





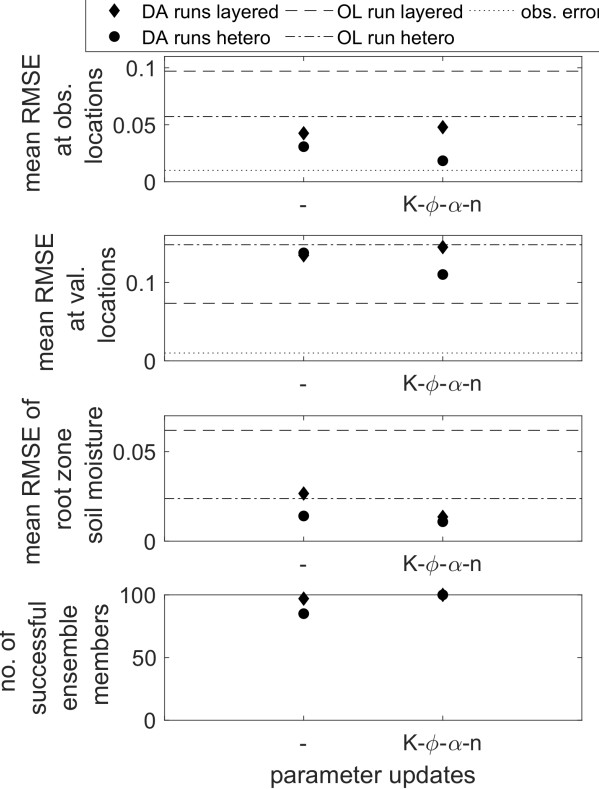

**Figure 18.** Spatially and temporally averaged $RMSE$ values at the observation and validation locations and for the averaged root zone soil moisture for the heterogeneous scenario using a heterogeneous and a simplified layered structure in the ensemble. The first entry (-) denotes the data assimilation run where only soil moisture is updated. The lower plot gives the number of converging ensemble members.

for two reasons. Firstly, the small spread impedes further updates as the simulated observations are given too much weight in Eq. 11. Therefore, the filter is not able to correct the states towards the observations once the ensemble spread has become too small. Secondly, the estimates seem more reliable than they actually are, which may lead to misinterpretations. At locations where one has no knowledge about the true state, it is then not possible to assess whether the states have taken a reliable value or not.


Therefore, it is important to prevent a too small ensemble spread. This can be achieved by thorough tuning of the filter properties, especially the dampening factor, the assimilation frequency and inflation. For test series involving multiple data assimilation runs, this is not feasible as these settings would have to be optimized for each individual run. Aside from that, different filter settings may decrease the comparability of the different assimilation runs. An optimized filter setup, that would

e.g. impede a strong spread reduction, would not change the conclusions of these experiments but would make it difficult to identify true and spurious correlations, which are relevant for the analysis of the different methods. Hence, we keep the same


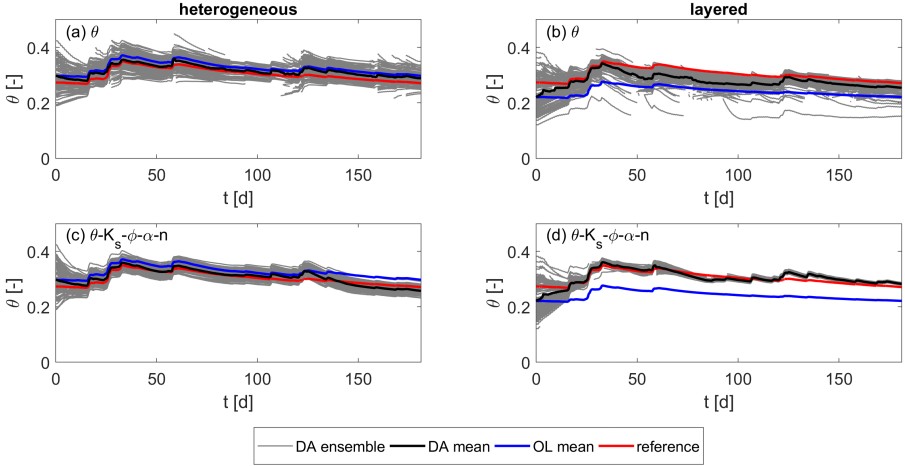

**Figure 19.** Root zone soil moisture over time for the heterogeneous scenario. *Left*: results using a heterogeneous structure in the ensemble; *right*: results using a simplified layered soil in the ensemble. Plots (a) and (b): updates of soil moisture only; plots (c) and (d): joint updates of soil moisture and all parameters.

settings for all runs, even though they may not be optimal in some cases. Yet, we want to stress that this applies only to the testing of data assimilation while a rigorous tuning for the assimilation in operational models is indispensable.

One concern when performing parameter updates, and in particular when including the parameter values of each grid cell in
the augmented state vector to resolve the heterogeneous structure, is that this increases the required computational resources. This is only partly true. Of course, larger matrices need to be handled, which can be either accomplished by using more work space (if available) or by a parallelization on multiple cores. As the large number of model realizations in the ensemble is anyhow often run in parallel mode, the latter option of handling the matrices would not require any additional resources, here. In terms of run times, the parameter updates have revealed a positive influence. For unsaturated flow problems, convergence
issues in single members of the ensemble are the main cause for long run times. As we have seen throughout our experiments, the parameter updates increase the numerical stability of the ensemble which actually reduces the run times.

Thus, there is no real drawback of applying parameter updates. Quite the contrary, updating the soil hydraulic parameters improves the soil moisture estimates in the homogeneous but also in the heterogeneous case notably. In the homogeneous scenario, the update of the van Genuchten model parameters turned out to be crucial for the numerical stability. In the het-
erogeneous test case this trend could not be confirmed. In Brandhorst et al. (2017) it was shown for the one-dimensional case that the influence of updating the individual parameters depends on the soil type and flow conditions. For example, porosity is particularly important when the soil is (almost) saturated and the observed soil moisture is $\theta \approx \theta_{sat} \approx \phi$. As the heterogeneous subsurface contains a larger spectrum of parameter values, a distinct influence of a specific parameter is therefore not to be expected, which is confirmed by the evaluation of the $RMSE$ of soil moisture shown in Fig. 10. For both comparisons (test
series one and two), updating multiple parameters leads to the best soil moisture estimates, both at the observation and at





the validation locations. According to our results, the update of porosity and the van Genuchten model parameters have the strongest impact on improving predictions of soil moisture. In Brandhorst et al. (2017) updating all four parameters led to the best results without exception. In the 3D heterogeneous setting considered here, this conclusion cannot be drawn so strictly. There were exceptions. However, updating all parameters always led to the best or second best results, such that it is advisable
to include all parameters in the updates.

In the heterogeneous test case, the importance of using a stochastic model becomes apparent. Due to the small correlations of parameters and states at other than the observation locations, the former are hardly updated and the initial ensemble spread is mostly maintained. The estimates at those locations are therefore rather poor. This reflects, however, the reality as at these locations there is really no information and they differ from the locations where observations are available. The large ensemble
spread indicates the high uncertainty at these locations making clear that estimates there are not reliable. A deterministic model does not quantify model uncertainty and would claim the wrong estimates to be correct. To improve the estimates, more information for the assimilation would be needed, e.g. in terms of more observations or remote sensing data.

In contrast, when a simplified layered soil structure in the ensemble is applied (test series three), the data assimilation fails entirely to improve the estimates aside the observation locations and leads to filter divergence. The ensemble spread is reduced
a lot leaving the model overconfident, whereas less accurate estimates in the heterogeneous ensemble setting go along with larger ensemble spreads that properly account for the remaining model uncertainty. At the observation locations, the estimates for the layered ensemble setting are also not as accurate as when the heterogeneous structure is resolved.

However, when estimating a spatially averaged quantity, in this case the spatially averaged root zone soil moisture, the accuracy when using the simplified soil structure is almost as good as when the fully heterogeneous structure is used. This
applies only when the parameters are updated. Updating only the states, the accuracy of the heterogeneous ensemble is clearly better.

From this one can summarize that one cannot obtain decent estimates of point values when applying a simplified soil structure, but it is possible to give decent estimates of cumulative values. In this case, again, the importance of parameter udpates becomes evident. Yet, this is supposed to work only if the observations are taken at locations where the parameter
values are somewhat representative for the mean parameter value of the domain. If the parameter values at these locations are in the extreme ranges of the parameter distributions, the estimation of cumulative values may fail, too. Here, we need to point out that the soil structure in the heterogeneous ensemble is also not exactly resolved. While the position of the interface between the layers is assumed to be known, the structure within the layers is not prescribed. The information is contained indirectly in the ensemble as the parameter fields of the individual ensemble members are created using the true correlation
lengths and almost correct correlations among the parameters. Yet, the correct spatial structure is not contained, neither in the individual parameters fields nor in the initial guesses. This can be intrepreted as applying a finer resolution of the heterogeneous layers with more degrees of freedom than in the reference soil whereas assigning a layered soil structure means the contrary. Prescribing the correct soil structure could possibly improve the estimates even more. As it is hard to obtain this information in the field, such a setting would be rather unrealistic, though. On the other hand, prescribing a wrong soil structure could lead to
worse estimates, as Erdal et al. (2014) found out for one-dimensional flow problems. Resolving heterogeneity in the ensemble





used for data assimilation is thus recommended, assigning information about the statistical properties of the heterogeneity that is available.

## 5    Conclusions

In this study, the ensemble Kalman filter was applied to a three-dimensional hillslope model to assimilate soil moisture. The
augmented state vector approach was used to investigate the influence of parameter updates on the soil moisture estimates. To this purpose two reference models were created, one with a homogeneous soil and the other one with two heterogeneous soil layers. These models provided synthetic observations for the assimilation and validation of the data assimilation runs.

A previous sensitivity analysis revealed the saturated hydraulic conductivity, porosity and the van Genuchten model parameters $\alpha$ and $n$ to be the most sensitive parameters with respect to soil moisture while the remaining parameters had a negligible
influence. An ensemble was generated for each reference model by perturbing the four sensitive parameters representing the uncertainty of these parameters. Then, a data assimilation run was performed for each possible combination of parameter updates to investigate the impact of the individual parameter updates on the soil moisture estimates. It was shown that for both scenarios, homogeneous and heterogeneous, the joint update of states (soil moisture) and the uncertain parameters improved the soil moisture estimates compared to runs without parameter updates. While updating the saturated hydraulic conductivity
turned out to be less important, the update of porosity and the van Genuchten model parameters were essential. Furthermore, the parameter updates improved the numerical stability of the ensemble resulting in a reduction of run time and consumed computational resources. It was further shown that a simplified representation of the heterogeneous soil structure leads to significantly worse estimates of local soil moisture values and filter divergence while it gave comparable results for estimates of averaged soil moisture when including parameter updates. Ignoring heterogeneous structures in data assimilation is therefore
only recommended if the aim of the model is to estimate cumulative quantities.

One issue that we encountered is that the improvement by the filter updates in heterogeneous soils is mostly limited to the observation locations and a small area around them. Estimates at more distant locations are still highly uncertain after the assimilation. More information is needed to overcome this problem. This can be achieved by using a denser measurement network. However, it is hardly feasible to install a monitoring network with the required density in a real field application.
Instead, the additional assimilation of remotely sensed data or observations from cosmic ray probes can be an option. Besides, the studies by e.g. Shi et al. (2015) and Zhang et al. (2018) indicate that additional measurements of groundwater level may help improve the soil moisture estimates.

Generally, the present study has shown that whenever the soil structure can be represented accurately in the ensemble (as e.g. in homogeneous soils), parameter updates are able to improve state estimates with optimally conditioned parameter estimates
reducing the model error caused by parameter uncertainty significantly. Yet, soil heterogeneity produces additional uncertainty in the model which needs to be accounted for. In this work, this was done by updating the fully heterogeneous parameter fields. Thus, the assimilation can reduce the model error caused by the soil heterogeneity as much as the observations allow for. By applying a simplified soil structure, this error can only be reduced to a very limited extent due to the insufficient degrees


of freedom in the ensemble. Besides, at some point this reduction can most likely not be further improved by assimilating more observations. Localization of the parameter updates, which in principle adds soil heterogeneity to the ensemble, may be beneficial in such cases.

Another open question remains that is related to other error sources in the model. The key message of this work regarding parameter uncertainty is: Take the whole lot, i.e. all sensitive parameters and the full heterogeneous soil structure. Using real data, uncertainties may also arise from the boundary conditions and the model error. How to optimally handle these errors and uncertainties is not yet thouroughly examined.

*Code availability.* TerrSysMP including its interface to PDAF is freely available at https://github.com/HPSCTerrSys/TSMP. PDAF is provided after registration on https://pdaf.awi.de/register/index.php.

*Author contributions.* Simulations and code enhancements were performed by NB. IN acquired the funding. Both authors contributed to the design of the experiments, the analysis of the results and writing the paper.

*Competing interests.* IN is a member of the editorial board of the journal.

*Acknowledgements.* Computing time has been provided by the Jülich Supercomputing Centre (http://www.fz-juelich.de).



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
