# Peer review of "Impact of parameter updates on soil moisture assimilation in a 3D heterogeneous hillslope model"

_Hydrology and Earth System Sciences, 2022_

## Referee Comment (RC1)

Review of the manuscript Impact of parameter updates on soil moisture assimilation in a 3D heterogeneous hillslope model by Natascha Brandhorst and Insa Neuweiler

**Summary**

The manuscript is well structured and written. Many studies have already been conducted on the topic, but I believe the present study provides some new insights. I have listed below only one general comment that would require additional analyses and some minor comments that should be considered to improve some descriptions and to strengthen the discussion. After that, the manuscript can be considered in my opinion for publication.

**General main comment**

[1] The effect of the DA in the validation points are generally poor. There is no specific information on the correlation length (L) of the generated random fields (L336) but I hypothesize that L is lower than the distance (d) between the location of the assimilated observation and validation observation. Thus, it should be interesting to see at which distance (d) DA improves the estimated soil moisture. There is no the need to run any other simulation but rather to calculate, e.g., RMSE at increasing d. It should be interesting to see and discuss if/when d > L the effect of DA is poor. On the one hand, this would help in defining where to install point-scale measurements. On the other hand, this would support the discussion of the low representativeness of point-scale measurements and the need for alternative soil moisture observations (e.g., L625).

**Specific comments in order of appearance (Line number L)**

L37. The problem of non-uniqueness due to insufficient observations is identified also in the present study. Moreover, however, the limited representativeness of point-scale soil moisture measurements is also highlighted. This could also be discussed in the conclusions of the present study.

L85. Due to the limited improvements of the DA when more realistic heterogenous cases are performed, I wonder if the simplified approaches proposed in literature (by the application of Miller scaling (Bauser et al., 2020) or global calibration coefficients (Shi et al., 2014)), might be preferable. I'm not asking to conduct any additional simulations or DA tests but these approaches could be further recall in the discussion and conclusions, i.e., what do the Authors think about using these approaches in the lights of the results obtained in the present study?

L233. A few details about the high performance computer and the computational resources used for these tests might be useful to highlight the effort for performing the simulations in the present study.

L240. Evaporation is prescribed and this might be one reason, in my opinion, of the instability of the simulations. If this is the case, I suggest the Authors extending the discussion based on that (e.g. at L385; at L431).

L233. Please specify here the thickness of each soil layer.

L246. If possible, please justify why you have used 181 days.

L250-252. The Authors well acknowledge that the experiments have been conducted eliminating some unwanted sources of uncertainty. Thus, it should be argued that, in real test cases, the results could be even worse than the one presented here. I would recall this aspect in the discussion and conclusion.

L291. How soil set-up is created (the random fields) is not well described. Information is reported only later (L335-338) but without information of the correlation length. I believe these are important details as different results can be obtained with different set-up. Thus, this information should also be presented at the beginning of the section.

L339. I suggest adding here a title "3.4 Performance metrices". I would then move the title "4. Results and discussion" before L354.

L420-421. Correlations between parameters and states at the validation locations and the observations are too small to induce an update of the former. This in my opinion could be related to the correlation length of the random field (L). See general comment #1 above. If this is the case, the discussion should be extended accordingly.

Figure 9. Not sure if I missed something, but I did not get why reference values are not plotted here. If possible, I would also add these lines (as red lines in figure 6). Discussion should be extended accordingly at L419.

L336-341. I would move NMD description to the method section after the description of the RMSE (i.e., L353). All together this sub-section should be named e.g., "3.4 Performance metrices"

L525. Point instead of colon?

L588. The term cumulative value could be in my opinion misleading. Please specify, e.g., variance of the field, cumulative density functions etc.

L614-615. Updating the saturated hydraulic conductivity turned out to be less important. This might be related to the upper boundary condition used (i.e., figure 2) while the results could be different when other hydrological conditions are prescribed. If this is the case, I would extend the discussion accordingly.

L620. Cumulative quantities. Please be more precise (see also comment above L588).

L625. Here I assume that you mean soil moisture observation from remote sensing and cosmic ray neutron probe. I would rephrase to be more precise.

---

## Author Comment (AC1)

Review of the manuscript Impact of parameter updates on soil moisture assimilation in a 3D heterogeneous hillslope model by Natascha Brandhorst and Insa Neuweiler

**Summary**

The manuscript is well structured and written. Many studies have already been conducted on the topic, but I believe the present study provides some new insights. I have listed below only one general comment that would require additional analyses and some minor comments that should be considered to improve some descriptions and to strengthen the discussion. After that, the manuscript can be considered in my opinion for publication.

We thank the reviewer for the effort and time to revise our manuscript and for the positive rating. In the following, we want to respond in detail to the constructive comments he/she provided.

**General main comment**

[1] The effect of the DA in the validation points are generally poor. There is no specific information on the correlation length (L) of the generated random fields (L336) but I hypothesize that L is lower than the distance (d) between the location of the assimilated observation and validation observation. Thus, it should be interesting to see at which distance (d) DA improves the estimated soil moisture. There is no the need to run any other simulation but rather to calculate, e.g., RMSE at increasing d. It should be interesting to see and discuss if/when d > L the effect of DA is poor. On the one hand, this would help in defining where to install point-scale measurements. On the other hand, this would support the discussion of the low representativeness of point-scale measurements and the need for alternative soil moisture observations (e.g., L625).

This is a very interesting suggestion. The horizontal correlation length in the model is L=2m and thus, as the reviewer noticed correctly, much smaller than the distance d to the validation points. We will perform such an analysis and include it in the revised manuscript.

**Specific comments in order of appearance (Line number L)**

L37. The problem of non-uniqueness due to insufficient observations is identified also in the present study. Moreover, however, the limited representativeness of point-scale soil moisture measurements is also highlighted. This could also be discussed in the conclusions of the present study.

This is a good suggestion. We will include a discussion on this issue in the conclusive part of the manuscript.

L85. Due to the limited improvements of the DA when more realistic heterogenous cases are performed, I wonder if the simplified approaches proposed in literature (by the application of Miller scaling (Bauser et al., 2020) or global calibration coefficients (Shi et al., 2014)), might be preferable. I'm not asking to conduct any additional simulations or DA tests but these approaches could be further recall in the discussion and conclusions, i.e., what do the Authors think about using these approaches in the lights of the results obtained in the present study?

This is a valid question. In our opinion, any additional constraints, that are imposed on the model, hinder the assimilation from reaching the optimal solution conditioned on the available observations. Simplified approaches are such constraints as they decrease the degrees of freedom of the data assimilation. Our approach to use a simplified layered soil structure is one example, although of course in a much stronger manner. There, we have seen that the filter performance is degraded and suppose a similar, yet less pronounced,

effect when using other simplifying approaches as e.g. Miller scaling or global calibration coefficients. We will add a paragraph regarding these simplified approaches in the discussion part of the manuscript.

L233. A few details about the high performance computer and the computational resources used for these tests might be useful to highlight the effort for performing the simulations in the present study.

We agree that this information is missing and will include it in the revised manuscript.

L240. Evaporation is prescribed and this might be one reason, in my opinion, of the instability of the simulations. If this is the case, I suggest the Authors extending the discussion based on that (e.g. at L385; at L431).

The reviewer is completely right here. The prescription of the evaporation flux, which does not consider the available water content in the upper soil, causes numerical instabilities. By assigning a less conductive soil layer with reduced spatial variability, a lid is kept on these instabilities. Alternatively, a moisture-dependent evaporation flux could have been implemented. Yet, this is only one reason for the numerical issues. These occur just as often during precipitation events as during evaporation events. Furthermore, there is no clear trend in the parameter combinations leading to numerical issues that allow for reliable conclusions. We will mention the prescribed evaporation as one reason for the numerical instabilities at the respective parts in the manuscript, but a more profound discussion would require additional testing.

L233. Please specify here the thickness of each soil layer.

We will do that.

L246. If possible, please justify why you have used 181 days.

We used a times series starting from the 1$^{st}$ of January and ending at the 30$^{st}$ of June. This sums up to 181 days. We will add this information.

L250-252. The Authors well acknowledge that the experiments have been conducted eliminating some unwanted sources of uncertainty. Thus, it should be argued that, in real test cases, the results could be even worse than the one presented here. I would recall this aspect in the discussion and conclusion.

Yes, this is to be expected and we will include this aspect in the discussion and conclusion part of the manuscript.

L291. How soil set-up is created (the random fields) is not well described. Information is reported only later (L335-338) but without information of the correlation length. I believe these are important details as different results can be obtained with different set-up. Thus, this information should also be presented at the beginning of the section.

We agree that this information should be given right at the beginning of this section. We will move lines 335-338 to the beginning and specify the used correlation length.

L339. I suggest adding here a title "3.4 Performance metrices". I would then move the title "4. Results and discussion" before L354.

This is a good idea and would increase the readability of the manuscript. We will change the title and add the performance matrices regarding the parameter estimates in l.437-441 and l.457-459 to this subsection, too.

L420-421. Correlations between parameters and states at the validation locations and the observations are too small to induce an update of the former. This in my opinion could be related to the correlation length of the random field (L). See general comment #1 above. If this is the case, the discussion should be extended accordingly.

As mentioned in our answer to general comment #1, we agree with the reviewer and will revise the manuscript accordingly.

Figure 9. Not sure if I missed something, but I did not get why reference values are not plotted here. If possible, I would also add these lines (as red lines in figure 6). Discussion should be extended accordingly at L419.

We can add the reference parameter distribution to the figure. This is not done in the original version because the message of the figure is the difference in ensemble spread and not the bias compared to the reference distribution. Yet, we understand that this could be an interesting information and will include it.

L336-341. I would move NMD description to the method section after the description of the RMSE (i.e., L353). All together this sub-section should be named e.g., "3.4 Performance metrices"

As stated in our answer to the comment on L339, we will do this. The same applies for the description of the NMV (l.457-459).

L525. Point instead of colon?

Yes, we agree that a point would fit better here and will change this.

L588. The term cumulative value could be in my opinion misleading. Please specify, e.g., variance of the field, cumulative density functions etc.

We mean spatially cumulative values. In this case, the spatial mean is taken as example. We will make this clearer.

L614-615. Updating the saturated hydraulic conductivity turned out to be less important. This might be related to the upper boundary condition used (i.e., figure 2) while the results could be different when other hydrological conditions are prescribed. If this is the case, I would extend the discussion accordingly.

No, quite the contrary, in combination with this boundary condition, we found that the saturated hydraulic conductivity is quite important. In other setups (1d experiments in Brandhorst et al., 2018) we saw a clear influence of updating the saturated hydraulic conductivity while also prescribing the evaporation flux. So there must be another reason for its update being less influential here, but as we could only guess here, we prefer not to extend the discussion into this direction and cannot say more than that this must be case-specific.

L620. Cumulative quantities. Please be more precise (see also comment above L588).

We mean spatially cumulative values and refer to our answer to the comment on L588.

L625. Here I assume that you mean soil moisture observation from remote sensing and cosmic ray neutron probe. I would rephrase to be more precise.

We agree that this formulation is misleading and will rephrase the sentence.

---

## Author Response (AR1)

**Response to review #1**

Review of the manuscript Impact of parameter updates on soil moisture assimilation in a 3D heterogeneous hillslope model by Natascha Brandhorst and Insa Neuweiler

**Summary**

The manuscript is well structured and written. Many studies have already been conducted on the topic, but I believe the present study provides some new insights. I have listed below only one general comment that would require additional analyses and some minor comments that should be considered to improve some descriptions and to strengthen the discussion. After that, the manuscript can be considered in my opinion for publication.

We thank the reviewer for the effort and time to revise our manuscript and for the positive rating. In the following, we want to respond in detail to the constructive comments he/she provided.

**General main comment**

[1] The effect of the DA in the validation points are generally poor. There is no specific information on the correlation length (L) of the generated random fields (L336) but I hypothesize that L is lower than the distance (d) between the location of the assimilated observation and validation observation. Thus, it should be interesting to see at which distance (d) DA improves the estimated soil moisture. There is no the need to run any other simulation but rather to calculate, e.g., RMSE at increasing d. It should be interesting to see and discuss if/when d > L the effect of DA is poor. On the one hand, this would help in defining where to install point-scale measurements. On the other hand, this would support the discussion of the low representativeness of point-scale measurements and the need for alternative soil moisture observations (e.g., L625).

This is a very interesting suggestion. The horizontal correlation length in the model is L=2m and thus, as the reviewer noticed correctly, much smaller than the distance d to the validation points. We performed such an analysis and included it in the revised manuscript (ll. 464 - 489). As we assigned different correlation lengths in the vertical and horizontal direction, we examined these directions separately. The results suggest that the area of influence, where the data assimilation leads to decent estimates, depends strongly on the parameter correlations and thus on the correlation lengths of the heterogeneous fields. Yet, a more profound analysis on this topic (considering more test cases with different correlation lengths) would be needed to draw thorough conclusions. Here, the investigation in the horizontal direction is very limited as the correlation length is only two times the grid size. Thus, neighboring cells are partly uncorrelated which results in increasing RMSE values already at d=1m. Thus, the range where a positive impact is expected (0<=d<1), cannot be analyzed because the grid is too coarse. In the vertical direction, on the contrary, the range d>0.1*L is missing where we would expect to see the transition between good and poor estimates.

**Specific comments in order of appearance (Line number L)**

L37. The problem of non-uniqueness due to insufficient observations is identified also in the present study. Moreover, however, the limited representativeness of point-scale soil moisture measurements is also highlighted. This could also be discussed in the conclusions of the present study.

This is a good suggestion. We added the phrase "given the very small radius of influence of point-scale soil moisture observations" in ll. 690 - 691.

L85. Due to the limited improvements of the DA when more realistic heterogenous cases are performed, I wonder if the simplified approaches proposed in literature (by the application of Miller scaling (Bauser et al., 2020) or global calibration coefficients (Shi et al., 2014)), might be preferable. I'm not asking to conduct any additional simulations or DA tests but these approaches could be further recall in the discussion and conclusions, i.e., what do the Authors think about using these approaches in the lights of the results obtained in the present study?

This is a valid question. In our opinion, any additional constraints, that are imposed on the model, hinder the assimilation from reaching the optimal solution conditioned on the available observations. Simplified approaches are such constraints as they decrease the degrees of freedom of the data assimilation. Our approach to use a simplified layered soil structure is one example, although of course in a much stronger manner. There, we have seen that the filter performance is degraded and suppose a similar, yet less pronounced, effect when using other simplifying approaches as e.g. Miller scaling or global calibration coefficients. We added several sentences discussing such approaches in ll. 638 - 648.

L233. A few details about the high performance computer and the computational resources used for these tests might be useful to highlight the effort for performing the simulations in the present study.

We included this information in ll. 232 -234.

L240. Evaporation is prescribed and this might be one reason, in my opinion, of the instability of the simulations. If this is the case, I suggest the Authors extending the discussion based on that (e.g. at L385; at L431).

The reviewer is completely right here. The prescription of the evaporation flux, which does not consider the available water content in the upper soil, causes numerical instabilities. By assigning a less conductive soil layer with reduced spatial variability, a lid is kept on these instabilities. Alternatively, a moisture-dependent evaporation flux could have been implemented. Yet, this is only one reason for the numerical issues. These occur just as often during precipitation events as during evaporation events. Furthermore, there is no clear trend in the parameter combinations leading to numerical issues that allow for reliable conclusions. We added a short part discussing the influence of this boundary condition on the numerical stability in ll. 244 – 251.

L233. Please specify here the thickness of each soil layer.

We added this information (l. 253).

L246. If possible, please justify why you have used 181 days.

We used a times series starting from the 1$^{st}$ of January and ending on the 30$^{st}$ of June. This sums up to 181 days. We motivated this choice in ll. 258 – 259: "this time series is long enough to allow for the filter to converge and to detect potential subsequent filter divergence."

L250-252. The Authors well acknowledge that the experiments have been conducted eliminating some unwanted sources of uncertainty. Thus, it should be argued that, in real test cases, the results could be even worse than the one presented here. I would recall this aspect in the discussion and conclusion.

Yes, this is to be expected and we mention this in the revised manuscript (ll. 630 – 631 and ll. 694 - 696).

L291. How soil set-up is created (the random fields) is not well described. Information is reported only later (L335-338) but without information of the correlation length. I believe these are important details as different results can be obtained with different set-up. Thus, this information should also be presented at the beginning of the section.

We agree that this information should be given earlier. We moved this part to lines 321 – 325 and specified the used correlation length.

L339. I suggest adding here a title "3.4 Performance metrices". I would then move the title "4. Results and discussion" before L354.

This is a good idea and would increase the readability of the manuscript. We changed the title and added the performance matrices regarding the parameter to this subsection, too (ll. 356 - 380).

L420-421. Correlations between parameters and states at the validation locations and the observations are too small to induce an update of the former. This in my opinion could be related to the correlation length of the random field (L). See general comment #1 above. If this is the case, the discussion should be extended accordingly.

As mentioned in our answer to general comment #1, we agree with the reviewer and we revised the manuscript accordingly. We included the analysis described in the general comment and referred to its findings in ll. 507 – 511, 533, 634 – 636 and 688.

Figure 9. Not sure if I missed something, but I did not get why reference values are not plotted here. If possible, I would also add these lines (as red lines in figure 6). Discussion should be extended accordingly at L419.

We added the reference parameter distribution to the figure. This was not done in the original version because the message of the figure is the difference in ensemble spread and not the bias compared to the reference distribution. Yet, we understand that this could be an interesting information and included it.

L336-341. I would move NMD description to the method section after the description of the RMSE (i.e., L353). All together this sub-section should be named e.g., "3.4 Performance metrices"

As stated in our answer to the comment on L339, we did this. The same applies for the description of the NMV (ll. 369 - 380).

L525. Point instead of colon?

Yes, we agree that a point would fit better here and changed this (l. 578).

L588. The term cumulative value could be in my opinion misleading. Please specify, e.g., variance of the field, cumulative density functions etc.

We mean spatially cumulative values. In this case, the spatial mean is taken as example. We rephrased to "spatially cumulative values" (ll. 654 and 686).

L614-615. Updating the saturated hydraulic conductivity turned out to be less important. This might be related to the upper boundary condition used (i.e., figure 2) while the results could

be different when other hydrological conditions are prescribed. If this is the case, I would extend the discussion accordingly.

No, quite the contrary, in combination with this boundary condition, we found that the saturated hydraulic conductivity is quite important. In other setups (1d experiments in Brandhorst et al., 2018) we saw a clear influence of updating the saturated hydraulic conductivity while also prescribing the evaporation flux. So there must be another reason for its update being less influential here, but as we could only guess here, we prefered not to extend the discussion into this direction and cannot say more than that this must be case-specific.

L620. Cumulative quantities. Please be more precise (see also comment above L588).

We mean spatially cumulative values and refer to our answer to the comment on L588.

L625. Here I assume that you mean soil moisture observation from remote sensing and cosmic ray neutron probe. I would rephrase to be more precise.

We agree that this formulation is misleading and rephrased the sentence accordingly (l. 692).

**Response to review #2**

Referee comment on "Impact of parameter updates on soil moisture assimilation in a 3D heterogeneous hillslope model" by Natascha Brandhorst and Insa Neuweiler, Hydrol. Earth Syst. Sci. Discuss., https://doi.org/10.5194/hess-2022-311-RC2, 2022

This is a very clearly written, well-constructed article that documents a numerical study of data assimilation using an integrated numerical model. I found this article scientifically interesting, as I also think the readership of HESS would. I recommend publication pending minor revisions. I have some general and specific comments listed below.

We thank the reviewer for the effort and time to revise our manuscript. We are pleased that the manuscript aroused the reviewer's interest and are thankful for the very positive rating and the constructive comments to which we will respond in the following.

General comments:

The integrated model used here incorporates overland flow. However, it's unclear what the role of overland flow was in the study as I don't believe it was used in the DA. Did the presence of overland runoff modify the soil moisture in some way (beyond acting as a boundary condition at the bottom of the hill slope)? Would the results be essentially the same with a Richards' only hill slope model?

The reviewer is right that overland flow was not used in the DA. In this model, overland flow only acts as a boundary condition, although there may be minor influences on soil moisture. Yet, we are convinced that the conclusions drawn in this work are not affected by the usage of overland flow and would have been the same when using a Richards' only model. Overland flow was included in the model as it allows for an investigation of the effect of streamflow data assimilation on soil moisture estimates. This was left for future work, though. Another reason to include overland flow was to avoid implementing an infiltration condition as a flux

boundary at the soil surface, where the total infiltration flux is imposed. This can lead to numerical problems, in particular under dry conditions. We added a small part discussing the influence of overland flow in the model in ll. 246 – 250.

The authors found that porosity was a particularly sensitive parameter in the DA. This makes sense, as they state (e.g. 615), as this limits the total amount of water available in the soil. However, this sensitivity is likely larger on the wet side of the soil moisture curve, on the dry side other parameters (processes) may play a larger role. The aridity of the simulations is driven by the meteorological forcing used in the experiment, did the authors consider the impact a different forcing dataset might have on these findings?

We agree with the reviewer that the wetness (or aridity) has an impact on the sensitivity of the individual parameters, especially porosity, and thus on their relevance in data assimilation. In previous one-dimensional experiments (Brandhorst et al., 2017), we had investigated the role of the different parameters in data assimilation under varying moisture conditions, although not generated by using different forcing data, but different reference soils. There, we saw a decreasing influence of porosity with increasing aridity. In the present model, we would expect a similar behavior albeit less pronounced since the heterogeneity of the soil always covers a larger range of the soil moisture curve. This is one reason while the concluding suggestion based on the experiments is to include all four parameters in the updates which was shown to be a good option for all test cases (including the 1D experiments conducted in Brandhorst et al., 2017). In this case one might include a less-sensitive parameter (as porosity in dry soils) but this would not have a negative effect on the assimilation. We added a part where this topic is discussed in ll. 620 – 627 and added the phrase "under the present flow conditions" in l. 680.

Specific comments:

Section 3.1.1: Are the random fields for ln(K), ln(alpha), phi and n spatially correlated? I realize they are correlated with each other, but are the fields disordered in space or correlated with some spatial correlation structure? There is a lot of evidence in the literature demonstrating the spatial correlation of random fields and this should be discussed in the manuscript. If the fields are correlated, I recommend including some discussion of the correlation model and associated parameters.

Yes, the fields are also spatially correlated. The horizontal correlation length is $L_H$=2m and the vertical correlation length $L_V$=4m, such that the vertical variability is negligible inside a soil layer. We agree that this is important information, which is missing in the manuscript, as the spatial correlation influences the data assimilation via the covariance matrix. We added it in l. 322.

Section 3.1/Table 2: For the numerical simulations were the random fields constrained in some way to prevent non-physical parameter values? Or even parameter values that would be outside the range of solution (VG parameters such as n that result in eqs 2, 3 being nondifferentiable).

This is a good remark. Yes, in addition to the requirements regarding mean, variance and correlations, we constrained the parameters to avoid unphysical or numerically difficult

values. In detail, the limits were: 0.14 < porosity < 0.76; 1 < VG-n < 5; 0.1 < VG-alpha < 57.5. For the saturated hydraulic conductivity, we did not use hard limits, but regulated the range by the applied variance and mean and then checked the values of the resulting fields on their physical plausibility. This is now mentioned in ll. 277 – 278 and 312 – 314.